# Dysferlin-mediated phosphatidylserine sorting engages macrophages in sarcolemma repair

Volker Middel[1], Lu Zhou[2,3], Masanari Takamiya[1], Tanja Beil[1], Maryam Shahid[1], Urmas Roostalu[4], Clemens Grabher[1], Sepand Rastegar[1], Markus Reischl[5], Gerd Ulrich Nienhaus[1,2,3,6] & Uwe Strähle[1]

Failure to repair the sarcolemma leads to muscle cell death, depletion of stem cells and myopathy. Hence, membrane lesions are instantly sealed by a repair patch consisting of lipids and proteins. It has remained elusive how this patch is removed to restore cell membrane integrity. Here we examine sarcolemmal repair in live zebrafish embryos by real-time imaging. Macrophages remove the patch. Phosphatidylserine (PS), an 'eat-me' signal for macrophages, is rapidly sorted from adjacent sarcolemma to the repair patch in a Dysferlin (Dysf) dependent process in zebrafish and human cells. A previously unrecognized arginine-rich motif in Dysf is crucial for PS accumulation. It carries mutations in patients presenting with limb-girdle muscular dystrophy 2B. This underscores the relevance of this sequence and uncovers a novel pathophysiological mechanism underlying this class of myopathies. Our data show that membrane repair is a multi-tiered process involving immediate, cell-intrinsic mechanisms as well as myofiber/macrophage interactions.

[1] Institute of Toxicology and Genetics, Karlsruhe Institute of Technology (KIT), PO Box 3640, 76021 Karlsruhe, Germany. [2] Institute of Applied Physics, Karlsruhe Institute of Technology (KIT), Wolfgang-Gaede-Straße 1, 76131 Karlsruhe, Germany. [3] Institute of Nanotechnology, Karlsruhe Institute of Technology (KIT), PO Box 3640, 76021 Karlsruhe, Germany. [4] Institute of Inflammation and Repair, The University of Manchester, Oxford Road, Manchester M13 9PL, UK. [5] Institute for Applied Computer Science, Karlsruhe Institute of Technology (KIT), PO Box 3640, 76021 Karlsruhe, Germany. [6] Department of Physics, University of Illinois at Urbana-Champaign, 1110 West Green Street, 61801 Urbana, Illinois, US. Correspondence and requests for materials should be addressed to U.S. (email: uwe.straehle@kit.edu).

Skeletal muscle cells are prone to plasma membrane lesions under physiological levels of mechanical stress. To prevent cell death and to avoid muscle regeneration with concomitant depletion of stem cell pools, lesions are rapidly sealed by a repair patch consisting of proteins and lipids[1,2]. The transmembrane protein Dysf plays a key role in restoring cell integrity. Humans with mutations in the *DYSF* gene acquire limb-girdle muscular dystrophy type 2B (LGMD2B), Miyoshi myopathy or distal myopathy with anterior tibialis onset[3,4]. Dysf binds phosphatidylserine (PS) in a $Ca^{2+}$-dependent manner through its C2-domains[5]. Mini-Dysf generated by $Ca^{2+}$-dependent calpain proteases and comprising the last two C2-domains of the full length mammalian protein appears to be the isoform of Dysf active in membrane repair[6]. In zebrafish, deletion of the C2 domains generated an isoform that is still able to accumulate at the site of lesion[7]. How Dysf precisely acts in membrane repair has remained elusive. Other players in membrane repair are annexins (AnxAs), which associate into multimeric complexes in a $Ca^{2+}$-dependent manner at the lesion[8–10]. These complexes are built by homo and heteromeric interactions of AnxAs and include also Dysf and possibly other repair proteins and lipids[7–10]. The distinct temporal order of proteins' arrival in the repair patch suggests that a specific structure of the protein-lipid matrix is required for its function[7]. Knock-down of Dysf or AnxA6 translation caused a malformed repair patch and double morphants even showed leakage of cytoplasmic components into the extracellular environment[7]. The origin of the membrane material forming the repair patch is unclear. Lysosomes as well as endosomes were suggested to contribute to membrane repair in mammalian cell systems[11,12]. However, none of the tested intracellular vesicles marked by Laptm4a, Lamp1, Lamp2, Rab1a, Rab5a, Rab6a, Rab7 and Rab27a contribute significantly to repair patch formation in zebrafish myofibers[7]. Although mechanisms like membrane shedding, exocytosis and endocytosis were discussed, a still unresolved issue is how the repair patch is removed to restore the plasma membrane[2].

We have employed live fluorescence imaging including single-molecule based techniques to study membrane repair in muscle cells in real time in zebrafish embryos and human cells. We provide evidence that macrophages remove the repair patch from damaged cells involving selective enrichment of PS at the lesion. Strikingly, Dysf mediates PS transport from adjacent sarcolemma to the repair patch through a five amino acid (AA)-motif close to its transmembrane (TM) domain, harbouring a point mutation in certain LGMD2B patients.

## Results

**Macrophages remove repair patch**. To assess up to which size a sarcolemmal lesion can still be repaired, we damaged the sarcolemma of single myofibers in the somitic musculature co-expressing plasmids encoding $Ca^{2+}$ sensing GCaMP5A (ref. 13) together with mOrange-tagged AnnexinA2a (AnxA2a-mO)[7]. After a transient increase of $Ca^{2+}$ ions at the lesion, a patch rapidly formed. All fibres with lesions $\leq 4\,\mu m$ survived membrane damage whereas lesions $\geq 4\,\mu m$ were associated with cell death (Supplementary Fig. 1a–g, Supplementary Movie 1, 2).

Frequently, we observed a motile cell appearing at the repair patch. We hypothesized neutrophils or macrophages were attracted to these sites and utilized *Tg(mpeg1:GFP)* and *Tg(lysC:dsRed)* marking macrophages[14] and neutrophils[15], respectively. The membranes of AnxA2a-mO expressing myofibers of transgenic fish were injured and leucocyte migration was monitored. Macrophages were recruited to injured myofibers (Fig. 1a–c, Supplementary Movie 3) in 50% of the cases

within $16.4 \pm 11.6\,min$ ($n=92$) after injury. In long-term (16–20 h) time-lapse experiments, 92% of the damaged myofibers had been visited by macrophages ($n=143$) (Supplementary Fig. 2a–c,g; Supplementary Movie 4). While myofibers with small lesions ($\leq 4\,\mu m$) attracted on average $1.29 \pm 0.2$ macrophages ($n=53$), about twice as many macrophages ($2.5 \pm 1.4$, $n=87$) were encountered at cells with lesions $\geq 4\,\mu m$. Macrophages were observed to ingest parts of the repair patch in surviving cells with lesion smaller than $4\,\mu m$ (Fig. 1a,c). In contrast, macrophages phagocytosed entire myofibers with lesions $>4\,\mu m$ (Supplementary Fig. 2a–c). Neutrophils were detected only infrequently (7%, $n=126$) at laser induced membrane lesions and did not participate in phagocytosis of dead myofibers. In contrast, sterility-compromised stabwounds inflicted by insertion of a glass needle into the somitic musculature attracted neutrophils in all cases ($n=23$) similar to tailfin cuts ($n=12$; Supplementary Fig. 2h).

To enquire if macrophages are crucial for repair patch removal, we employed an established morpholino knock-down strategy[16–18]. By triple injection of morpholinos directed against *gcsfr*, *pu1* and *irf8* mRNAs, we eliminated *Tg(mpeg1:GFP)* expressing macrophages. In controls (Fig. 1d, $n=37$), the repair patch was removed in 62% of the damaged cells, whereas, in all morphants (Fig. 1d, $n=25$, $P<10^{-8}$), the patch persisted beyond the observation time (20 h). Myofibers damaged beyond repair were not phagocytosed in the morphants (Supplementary Fig. 2d–g, $n=20$), whereas 92% were removed within 20 h in controls (Supplementary Fig. 2a–c, g, $n=12$; $P<10^{-8}$). We obtained the same results when we analysed a Crispr/Cas9 generated loss-of-function mutation in *irf8* (Fig. 1e) underscoring that the observed morpholino effects were specific. Taken together, these findings strongly suggest a crucial and novel role of macrophages for selectively removing the repair patch at membrane lesions.

**Fast and selective PS accumulation**. Exposure of PS to the extracellular space can trigger phagocytosis by macrophages[19]. To investigate, whether PS accumulates in the repair patch, we co-expressed the stereo-specific and $Ca^{2+}$-independent PS sensor LactAdherinC2:GFP (LactC2:GFP)[20] together with AnxA2a-mO. The PS sensor accumulated earlier than AnxA2a-mO (Fig. 2a, Supplementary Fig. 3a; Supplementary Movie 5). Dysf fused to mOrange (Dysf–mO)[7] accumulated at the same pace as LactC2:GFP (Fig. 2b, Supplementary Fig. 3b) or TopFluor-PS (ref. 21) at membrane lesions (Supplementary Fig. 3c). Clearly, PS is one key component of the early repair patch.

We next analysed the selectivity of PS accumulation. Membrane tethered CAAX-mCherry[22] accumulated more slowly (Fig. 2c, Supplementary Fig. 3d) compared with LactC2:GFP (Fig. 2c). Lyn-tailed mCherry[23], another membrane marker, showed no accumulation at the lesion (Fig. 2d, Supplementary Fig. 3e). We also tested lipid sensors recognizing phosphatidylinositol 4,5-bisphosphate (PI(4,5)P2, GFP-2 × PH(PLCδ))[24], phosphatidylinositol (3,4,5)-trisphosphate (PIP3; pEGFP::2FYVE–GFP)[25], phosphatidylinositol 3,4-bisphosphate (PI(3,4)P2, AKT-PH: EGFP)[25,26] and a fluorescently tagged cholesterol (BODIPY-cholesterol)[27]. Except for cholesterol (Fig. 2h, Supplementary Fig. 3i), none of the lipid markers accumulated at levels comparable to PS (Fig. 2e–g, Supplementary Fig. 3f–h).

To assess whether PS is presented to macrophages at the repair patch, we expressed the secreted PS sensor secA5-YFP (ref. 28) and injured non-expressing myofibers (Fig. 2i–l). Hence, secA5-YFP was exclusively supplied to the lesion patch from the extracellular space (Fig. 2j,k). secA5-YFP was enriched at the lesion (Fig. 2l, Supplementary Movie 6) demonstrating that PS is present at the extracellular surface of the lesion patch.

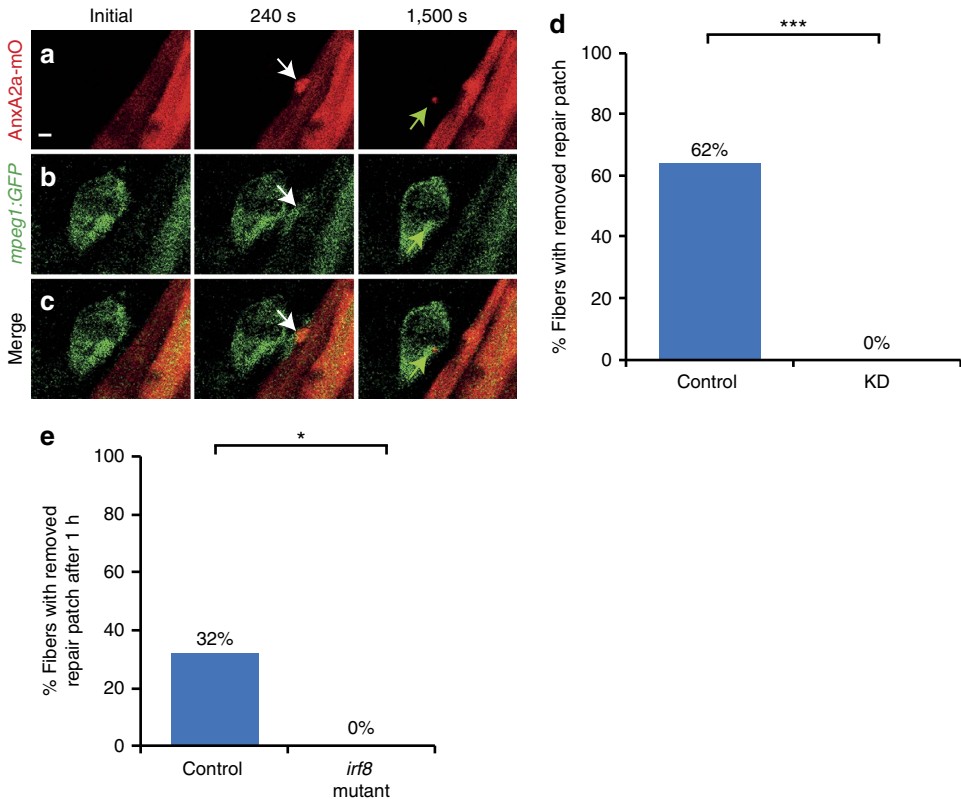

**Figure 1 | Macrophages remove repair patch.** (**a–c**) Repair patch (AnxA2a-mO, a, white arrow) and macrophages (*mpeg1:GFP*, b, white arrows) before (Initial), 240 s and 1,500 s after wounding. A macrophage ingests parts of repair patch (green arrows). (**c**) Merged views of **a,b**. Note the bleed-through of AnxA2a-mO into the GFP channel (**b**). (**d**) Repair patch removal in control (left, $n = 37$) and macrophage-depleted embryos (KD, right, $n = 25$, Fisher's exact test $P < 10^{-8}$) over 20 h. (**e**) Crispr/Cas9 knock-out of *irf8* impairs repair patch removal. In controls, the repair patch was removed in 32% of injured myofibers ($n = 19$) within one hour after injury. In the *irf8-KO* embryos the patch was present in all cases of injured myofibers examined ($n = 15$). This suggests that morpholino triple knock-down is inefficient and the same effect on macrophages can be achieved by elimination of *irf8* at the stage analysed. Significance was checked with Fishers exact test at $P < 0.05$ ($P = 0.023895$). Scale bar, 4 μm.

We reasoned that secA5-YFP masks PS at the lesion patch and thereby delays phagocytosis. Indeed, phagocytosis was delayed in embryos expressing secA5-YFP ($n = 19$; $349 \pm 241$ min) in comparison with Dysf–mO ($n = 23$; $108 \pm 104$ min; $P < 0.00001$) or AnxA2a-mO ($n = 22$; $124 \pm 79$ min; $P < 0.00001$) expressing fibres (Fig. 2m). Taken together, PS rapidly and selectively accumulates in the lesion patch, is presented to the outside, and masking of PS delays phagocytosis. All these observations clearly support the notion that PS marks the repair patch for phagocytosis by macrophages.

**Dysf is required for PS accumulation.** Since Dysf fragments and LactC2:GFP accumulated at similar speeds, we tested whether Dysf plays an active role in PS accumulation. We inflicted membrane wounds in myofibers expressing LactC2:GFP in combination with morpholino (MO) knock-down of endogenous Dysf[7]. Injecting a 5 bp mismatch *contr*-MO caused normal PS accumulation (Fig. 3a,c). Injection of *dysf*–MO, however, abolished PS recruitment (Fig. 3b,c, Supplementary Fig. 4a). Co-injection of *dysf–MO* together with *mOrange1–DysfC*[7] restored accumulation of PS (Fig. 3c, Supplementary Fig. 4a). AnxA6 is required for the formation of a tight lesion patch[7]. In *anxa6* morphants, PS accumulation was indistinguishable from that of controls (Fig. 3d, Supplementary Fig. 4b). Furthermore, *dysf*-mutant embryos, in contrast to wild-type siblings showed significantly reduced PS accumulation (Supplementary Fig. 4c–f).

Since a 74 AA C-terminal fragment of Dysf[7] was sufficient to restore PS translocation towards the lesion, we next asked which region in Dysf (Fig. 3e) mediates PS accumulation. Thus, we shortened the fragment further to a predicted amphipathic helix[29], residing N-terminally of and within the putative TM domain (Fig. 3e, zfWRRFK-TM-C). zfWRRFK-TM-C localized to the Z-lines and sarcolemma and translocated to the repair patch after membrane injury (Fig. 3f,h). Deletion of 14 AAs from the C-terminus (zfWRRFK-TM) did not affect accumulation (Fig. 3e,h, Supplementary Fig. 4g). However, removal of 5 AAs N-terminally led to mislocalization in intact cells and failure to accumulate (Fig. 3g,h, Supplementary Fig. 4g). Strikingly, PS accumulation was not rescued in *dysf*-morphants co-injected with *zfTM-C*, while *zfWRRFK-TM-C* rescued PS enrichment (Fig. 3i, Supplementary Fig. 4h), similar to *dysf*-mutant embryos (Supplementary Fig. 4i,j). In conclusion, this 5-AA motif which is not related to the reported PS binding C2-domains of Dysf (ref. 5) is essential for the selective accumulation of PS at the repair patch.

**Dysf mediates PS relocation.** To assess the dynamics of Dysf–mO, we employed fluorescence loss in photobleaching (FLIP). Dysf–mO expressing myofibers were imaged every second for 118 s under a spinning disk laser scanning microscope (Fig. 4a) and damaged by 405-nm laser irradiation. The light exposure also caused substantial bleaching at the site of lesion (Fig. 4a). As a control, we irradiated with 561 nm instead of 405 nm laser light, which only bleaches but does not damage the

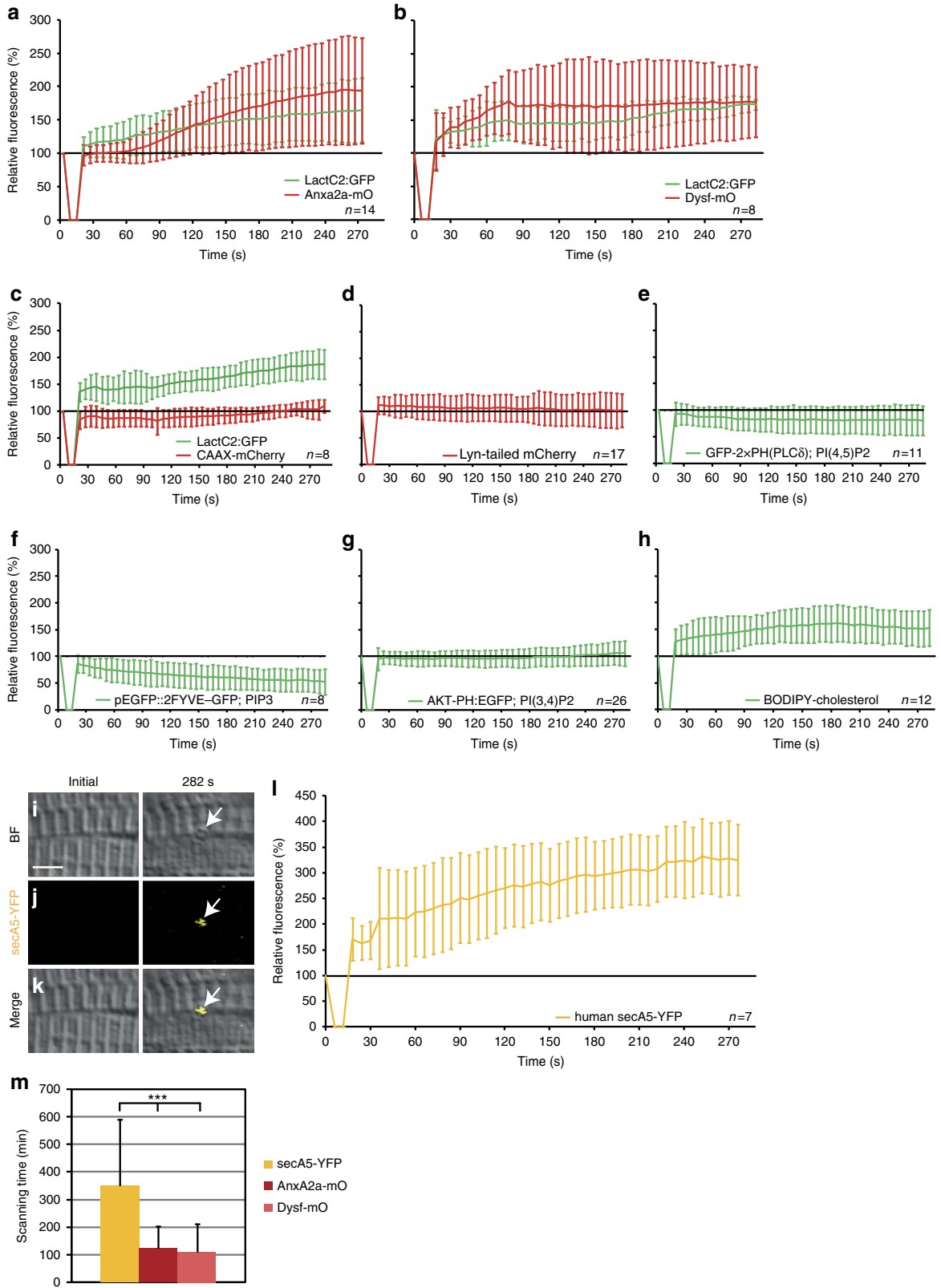

**Figure 2 | Phosphatidylserine (PS) is sorted to the repair patch.** (**a–h**) Real-time analysis of enrichment of PS (LactC2:GFP (**a–c**)), PI(4,5)P2 (GFP-2 × PH(PLCδ) (**e**)), PIP3 (pEGFP::2FYVE–GFP (**f**)), PI(3,4)P2 (AKT-PH:EGFP (**g**)) and cholesterol (BODIPY-cholesterol (**h**)), relative to AnxA2a-mO (**a**), Dysf–mO (**b**) and membrane markers CAAX-mCherry (**c**), Lyn-tailed mCherry (**d**). The reporter fluorescence is expressed as percentage (mean ± s.d.) relative to the level before injury. (**i–m**) PS is presented on the extracellular side of the repair patch. (**i–l**) Extracellularly supplied secA5-YFP was enriched at membrane on lesioning (**i–k**, arrows). secA5-YFP expressing myofibers are outside of the field of view. (**l**) Kinetics of secA5-YFP at the repair patch. (**m**) Macrophage scanning-time (+/− s.d.) of myofibers expressing secreted secA5-YFP (yellow, n = 19), AnxA2a-mO (dark red, n = 22) or Dysf–mO (light red, n = 23; Student t-test P < 0.001) Scale bar, 4 μm.

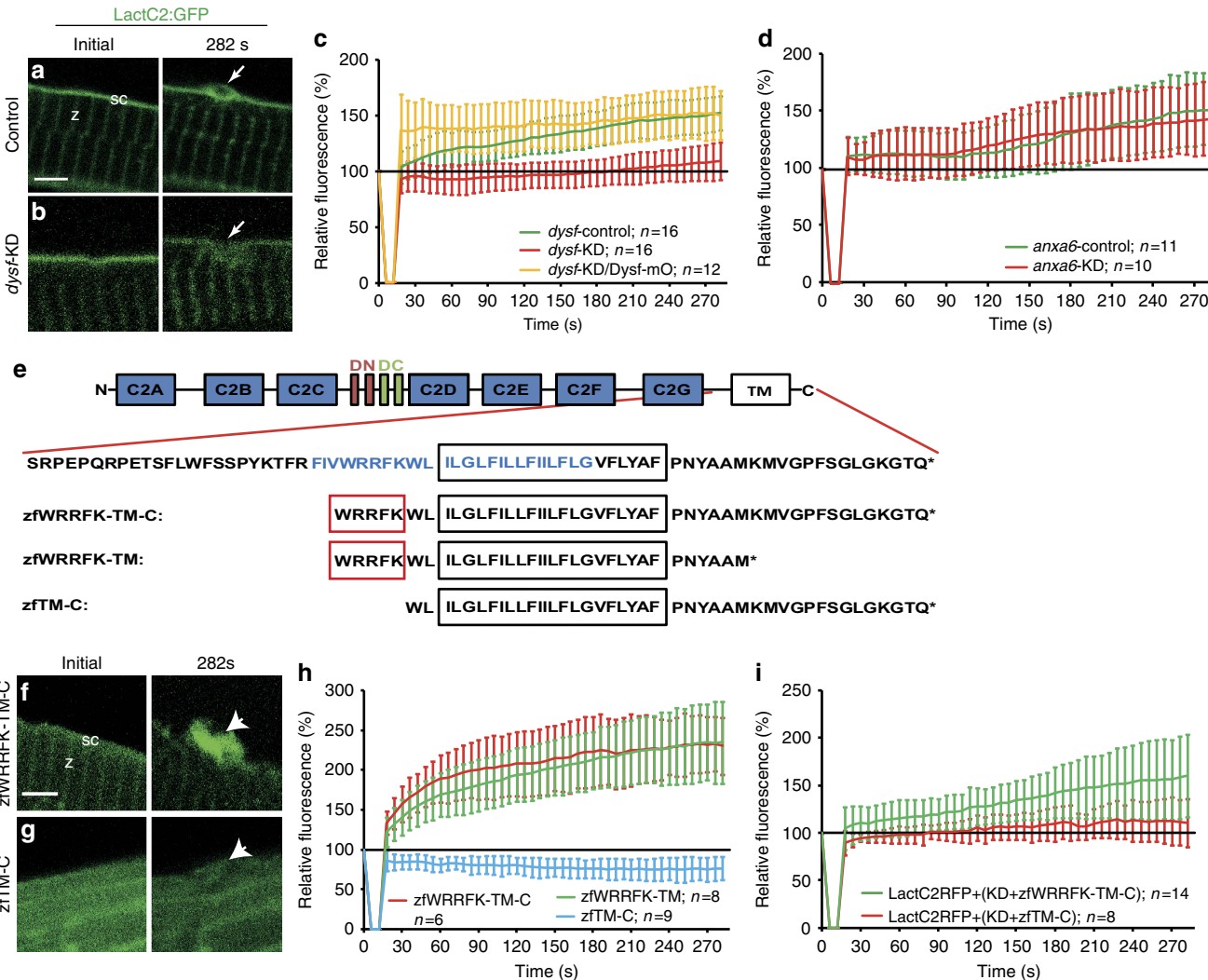

**Figure 3 | Five amino acid motif in Dysf is required for PS accumulation.** (**a,b**) Before wounding, LactC2:GFP (green) localized to the sarcolemma (sc; **a**) and the Z-line (z; a) both in control (**a**) and knock-down (*dysf*-KD; **b**) myofibers. On injury, LactC2:GFP accumulated in controls (**a**, arrow) but not in *dysf*-KD myofibers (**b**, arrow). (**c**) Kinetics of LactC2:GFP accumulation in control-KD (green), *dysf*-KD (red) and *dysf*-KD embryos co-injected with *mOrange1-DysfC* (Dysf–mO, yellow). Lack of PS accumulation in *dysf*-KD myofibers was rescued by Dysf–mO, translation of which is not inhibited by *dysf* morpholino. (**d**) Knock-down of *anxa6* (*anxa6*-KD) had no effect on PS accumulation (red) compared with controls (*anxa6-control*, green). (**e**) Domain structure of Dysf. (blue AA: predicted amphipathic helix). (**f,g**), On damage zfWRRFK-TM-C accumulates in the repair patch (f, arrow). No accumulation was observed for zfTM-C (**g**, arrow). (**h**) Accumulation kinetics of zfWRRFK-TM-C (red), zfWRRFK-TM (green) and zfTM-C (blue). A 5-AA motif (WRRFK, red box, **e**) is required for accumulation. (**i**) zfWRRFK-TM-C (red) but not zfTM-C (green) rescued PS accumulation in *dysf*-KD myofibers. In all charts, the change of fluorescence at the lesion is indicated as percentage relative to the undamaged state ± s.d.). Scale bars, 4 μm (**a,g–h**), 3 μm (**b**).

membrane (Fig. 4b). After local bleaching using 561-nm irradiation, we observed a fluorescence intensity decrease in the region of interest adjacent to the bleached site followed by slow recovery (Fig. 4c). In contrast, after local bleaching and membrane damage by 405-nm irradiation, the fluorescence intensity decreased faster and to a significantly lower level (Fig. 4c, Supplementary Fig. 5a). The faster kinetics and greater extent of fluorescence loss after membrane damage provide strong evidence that Dysf from adjacent, intact sarcolemmal regions is recruited to the site of lesion (Fig. 5a).

To directly visualize the movement of individual molecules to the wounded area, we performed super-resolution photoactivation localization microscopy, using green-to-red photoconversion of mEosFP (ref. 30). From these data, we analysed the displacements and speeds of $130 \pm 5$ single-molecule trajectories of mEosFP:zfWRRFK-TM-C along a line connecting the trajectory midpoint and the damage site in

regions 2–15 μm away from the wound (Fig. 4d). A parameter, $R_{proj.}$, was defined revealing random ($R_{proj.} = 1$) or net directional motion towards ($R_{proj.} > 1$) or away from ($R_{proj.} < 1$) the site of lesion (see supplementary experimental procedures). Figure 4e shows a super-resolved image of a mEosFP:zfWRRFK-TM-C labelled myofiber in zebrafish; the arrow indicates the site of damage. The resulting trajectories are shown in Fig. 4e. For zfWRRFK-TM-C, we observed a strong tendency to move towards the damage site ($R_{proj.} = 1.92$) and to accumulate at the membrane lesion (Fig. 4f). For the undamaged membrane region on the left (Fig. 4b), directional motion is absent ($R_{proj} = 1.04$), calculated with respect to a randomly chosen point (black dot). Control experiments with lipid-anchored CAAX-mEosFP did not display directional motion in the presence of damage (Fig. 4f, $P < 10^{-8}$), showing that directional motion of zfWRRFK-TM-C was not caused by overall membrane shifts towards the lesion.

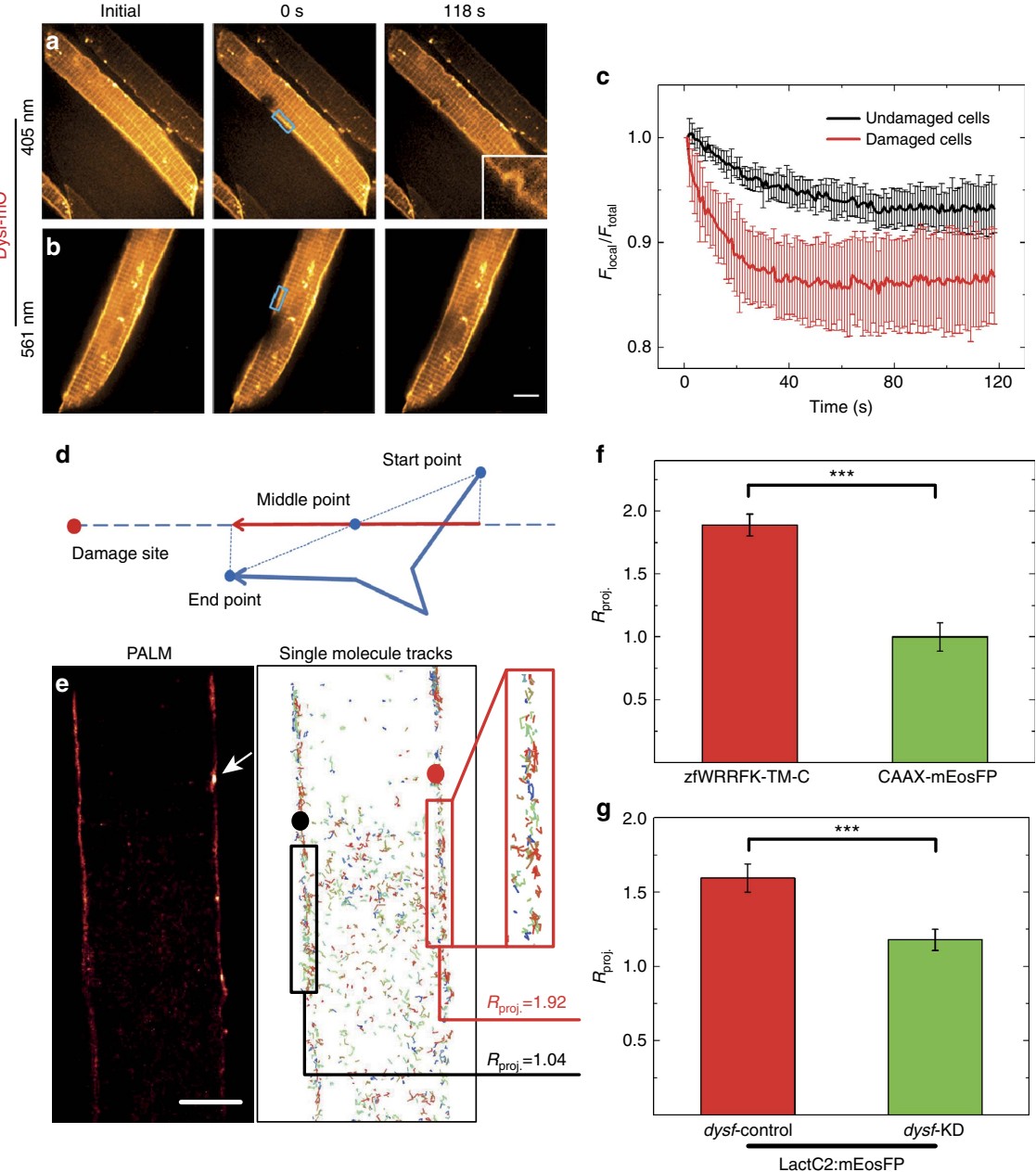

**Figure 4 | Dysf facilitates PS translocation to site of lesion.** (**a**–**c**) Fluorescence loss in photobleaching (FLIP) analysis on selected regions (blue boxes in **a**,**b**) of live Dysf–mO expressing myofibers. (**a**) Photobleaching and membrane damaging using 405 nm laser; (**b**) photobleaching only using 561 nm laser. (**c**) 561-nm irradiation results in an intensity decrease by 6% (black curve); 405-nm irradiation results in a more rapid decrease by 14% (red curve). (**d**–**g**) Single-molecule trajectory analysis of zfWRRFK-TM-C and controls after sarcolemmal damage. (**d**) Scheme showing the projection of a single-molecule trajectory onto the line connecting the midpoint with the site of damage. (**e**) Super-resolution localization image of a mEosFP:zfWRRFK-TM-C labelled myofiber. The arrow indicates the site of lesion. Single-molecules trajectories calculated from the image data in **e**. Molecules near the lesion (red box) show a high tendency ($R_{proj.} = 1.92$) to move towards the lesion (red dot). In the undamaged sarcolemma (left), trajectories (black box) did not show directed motion towards the black dot ($R_{proj.} = 1.04$). (**f**) mEosFP:zfWRRFK-TM-C moved towards the lesion (red column); the control CAAX-mEosFP did not (green column; $P < 10^{-8}$). (**g**) The PS sensor LactC2:mEosFP moved towards the lesion in *dysf*-contr-KD embryos (red column) but not in *dysf*-KD embryos (green column; Student *t*-test $P < 10^{-8}$). $n \geq 8$. Scale bar, 5 μm.

We performed additional single-molecule tracking experiments by using the PS sensor LactC2:mEosFP in *dysf* knock-down and control embryos. LactC2:mEosFP showed a much higher tendency to move towards the lesion in control than in *dysf* knock-down embryos (Fig. 4g, $P < 10^{-8}$), further supporting our claim that Dysf plays a decisive role in translocating PS within the membrane to the damage site.

**Dysf and PS accumulation require arginines.** Double arginine (RR) motifs have previously been implicated in $Ca^{2+}$-independent interaction with PS (ref. 31). Therefore, we tested the relevance of arginines within the WRRFK-motif (Fig. 3e) for accumulation of the Dysf and PS reporters. In contrast to zfWRRFK-TM-C, zfWAAFK-TM-C did not show any enrichment at the lesion (Fig. 5a, Supplementary Fig. 6a). Mutation of either one of the two arginines reduced translocation to the lesion (Fig. 5a,

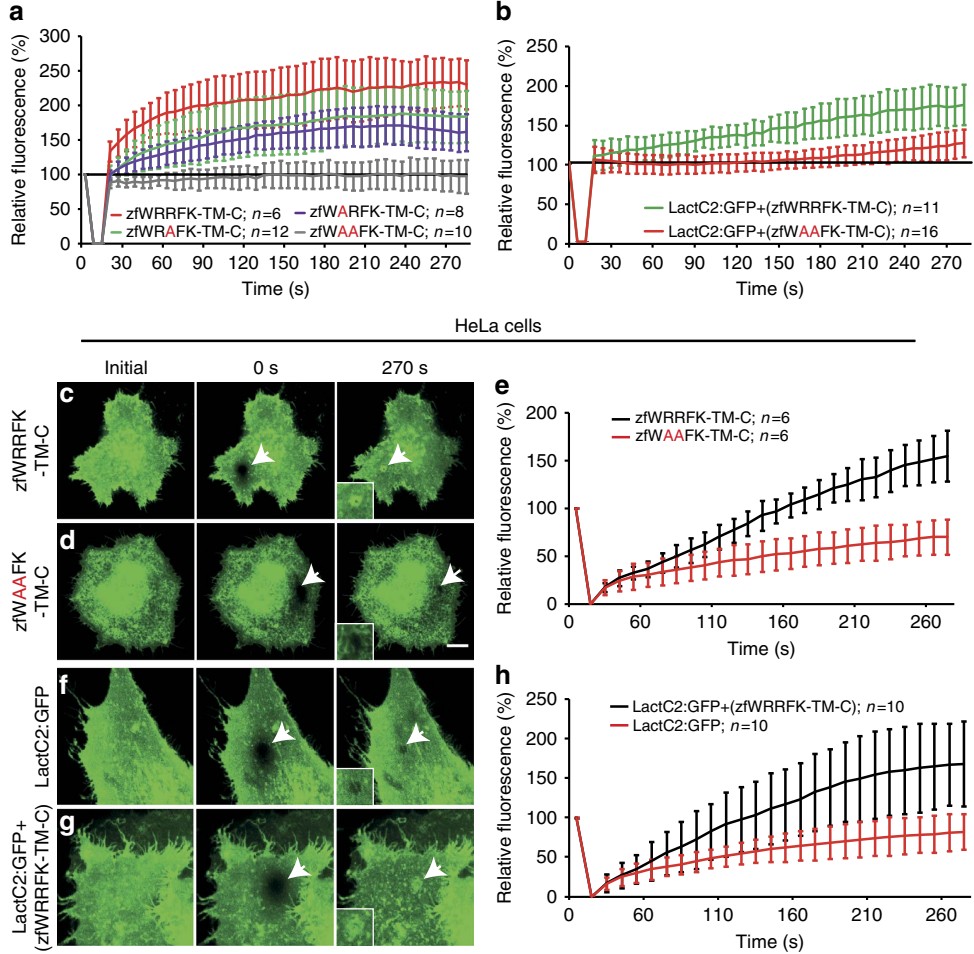

**Figure 5 | The WRRFK-motif is required for accumulation of Dysf and PS.** (**a**) After membrane damage, zfWRRFK-TM-C (red) but not zfWAAFK-TM-C (grey) accumulated rapidly at the lesion. The effect was reduced on exchange of the first (zfWARFK-TM-C; purple) or second arginine by alanine (zfWRAFK-TM-C; green). (**b**) PS (LactC2:GFP) enrichment in the repair patch of Dysf-KD myofibers co-expressing zfWRRFK-TM-C (green), but not of those co-expressing zfWAAFK-TM-C (red). (**c–e**) HeLa cells expressing zfWRRFK-TM-C were imaged before (**c**) and 0 s (**c**, arrow) and 270 s after membrane damage (**c**, arrows). zfWRRFK-TM-C markedly amassed at the lesion (**c**, arrow, inset; **e**, black), whereas zfWAAFK-TM-C showed only baseline fluorescence recovery after photobleaching (**d**, arrows, inset; **e**, red). (**f–h**) HeLa cells transfected with LactC2:GFP and imaged before (**f**) and 0 s (arrow) and 270 s after lesioning (arrow) showed no PS accumulation but only baseline recovery (**f**, arrow, inset; **h**, red); in the presence of zfWRRFK-TM-C, LactC2:GFP accumulated within the repair patch (**g**, arrows, inset; h, black). The data in (**a,b,e,h**) are given as mean ± s.e.m. ($n = >5$) and scaled such that 100% corresponds to fluorescence from the same area before damaging. Scale bars, 10 μm (**c,d**) and 12.86 μm (**f,g**).

Supplementary Fig. 6a), demonstrating that both contribute to Dysf accumulation (Fig. 5a).

To enquire whether mutating the RR motif affects PS enrichment at the lesion, we knocked-down *dysf* and expressed LactC2:GFP together with either zfWRRFK-TM-C or zfWAAFK-TM-C. While zfWRRFK-TM-C rescued PS accumulation, *dysf*-morphants expressing zfWAAFK-TM-C showed only partial rescue (Fig. 5b, Supplementary Fig. 6b).

**PS accumulation in human cells**. We tested whether zfWRRFK-TM-C accumulates in a heterologous system at injured cell membranes using human HeLa cells expressing very low levels of endogenous DYSF (ref. 32). Membrane damage in zfWRRFK-TM-C expressing HeLa cells (Fig. 5c) resulted in significant enrichment in the lesion patch (Fig. 5c,e), whereas zfWAAFK-TM-C showed no accumulation (Fig. 5d,e, Supplementary Fig. 6c). We examined whether the zebrafish Dysf fragment leads to enrichment of PS at the site of lesion in HeLa cells. Co-expression of LactC2:GFP and zfWRRFK-TM-C

led to significant accumulation of PS at membrane lesions (Fig. 5f–h, Supplementary Fig. 6d–f). Undifferentiated C2C12 myoblasts showed LactC2:GFP accumulation only when co-expressing zfWRRFK-TM-C (Supplementary Fig. 6g–j). In contrast, when C2C12 cells were differentiated to myotubes, which express high amounts of endogenous DYSF, LactC2:GFP showed enrichment at the lesion site in the absence of co-transfected zfWRRFK-TM-C (Supplementary Fig. 6k,l). Thus, the C-terminal fragment of zebrafish Dysf is sufficient to mediate PS accumulation at a membrane lesion in human cells, strongly suggesting that the underlying mechanisms are conserved between fish and humans.

**Dysferlinopathy caused by mutation of RR motif**. A sequence similar to the zebrafish motif WRRFK is present in human Dysf (Fig. 6a, WRRFR). Therefore, we tested whether human WRRFR-TM-C also would accumulate in zebrafish. Indeed, hWRRFR-TM-C showed high enrichment at the lesion (Fig. 6b,c). The double arginine mutant hWAAFR-TM-C showed

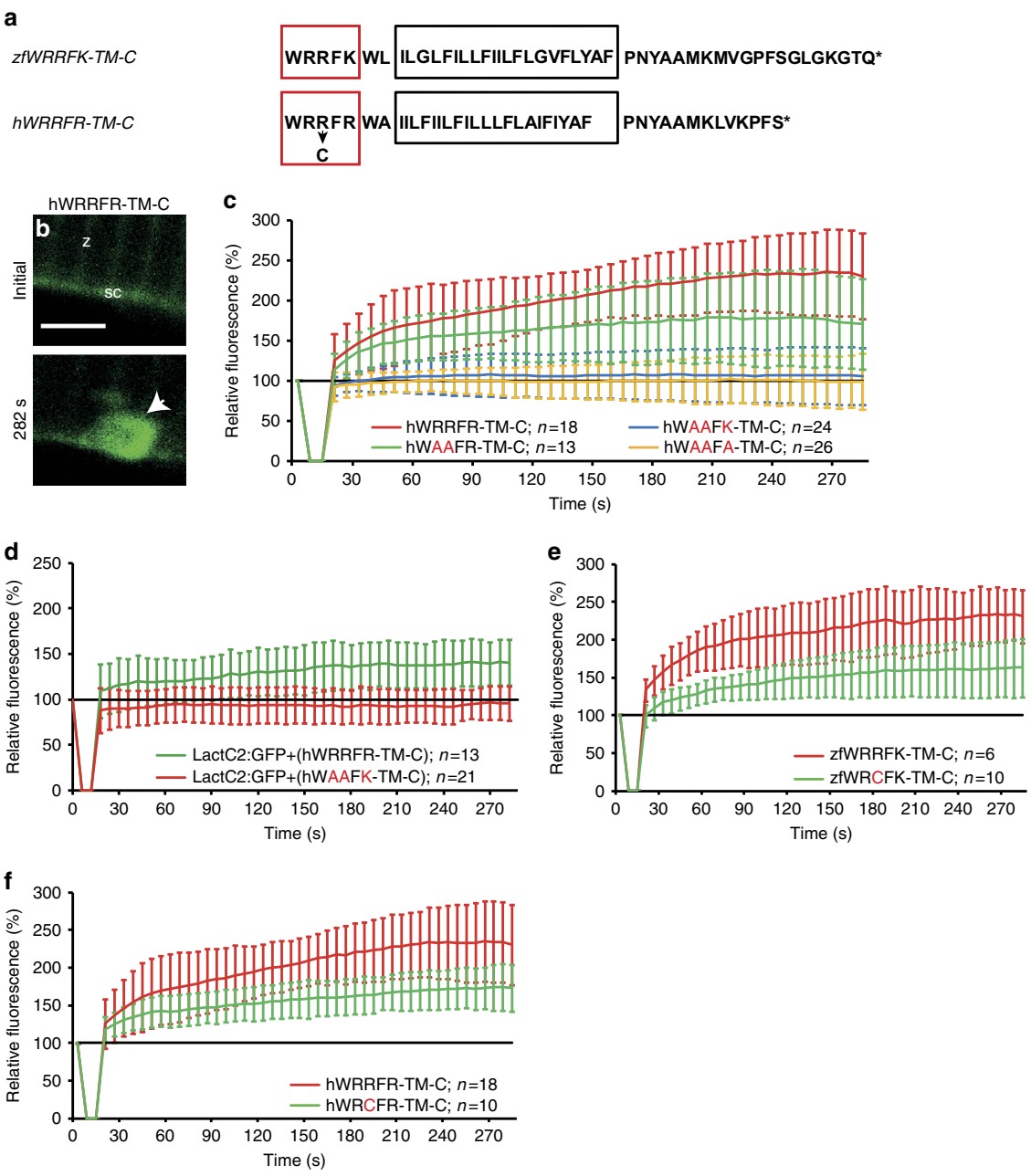

**Figure 6 | Arginine-rich motif is mutated in myopathy patients.** (**a**) Zebrafish zfWRRFK-TM-C and human hWRRFR-TM-C (red box arginine-rich motif; black box: TM domain, arrow: R2042C mutation). (**b,c**), hWRRFR-TM-C localized to the Z-line (z) and sarcolemma (sc) before damage (**b**). On damage, it accumulated at the lesion (**b**, arrow; **c**, red). In contrast, hWAAFK-TM-C (**c**, blue) and hWAAFA-TM-C (**c**, yellow) mutants did not accumulate and hWAAFR-TM-C caused significantly reduced accumulation (**c**, green). (**d**) LactC2:RFP accumulated in Dysf-KD myofibers when hWRRFR-TM-C (green) but not when mutant hWAAFK-TM-C (red). was co-expressed. (**e,f**) zfWRCFK-TM-C (**e**, green) and hWRCFR-TM-C (**f**, green), modelling the R2042C dysferlinopathic mutation showed significantly reduced accumulation (red). The fluorescence intensity of Dysf reporters at lesion relative to levels before damage is expressed as mean ± s.d. ($n \geq 6$). Note that the control data on hWRRFR-TM-C (**c,f**) are identical. Similarly, zfWRRFK-TM-C data (**e**) are also shown in Figs 3h and 5a. Scale bar, 4 μm.

reduced accumulation compared with hWRRFR-TM-C, yet, enrichment was not completely abolished, suggesting the third arginine may contribute to the activity (Fig. 6c, Supplementary Fig. 7a). On replacement of the third arginine by alanine (hWAAFA-TM-C) or lysine (hWAAFK-TM-C), accumulation was strongly impaired (Fig. 6c, Supplementary Fig. 7a).

Next, we knocked-down *dysf* and asked whether hWRRFR-TM-C rescued PS accumulation. Indeed, the PS sensor accumulated when hWRRFR-TM-C, but not when mutant hWAAFK-TM-C was co-expressed (Fig. 6d, Supplementary Fig. 7b).

Patients who presented with LGMD2B/MM were reported to have a point mutation in one of the arginines (R2042C, WRCFR)[33]. Thus, we mutated the corresponding arginine to cysteine (zfWRCFK-TM-C, hWRCFR-TM-C). In agreement with our mutant analysis (Figs 5a and 6c), this modification did not totally abolish, but significantly reduced accumulation of the zebrafish and the human DYSF reporter (Fig. 6e,f, Supplementary Fig. 7c–d). Hence, a likely cause of LGMD2B in these patients is inefficient DYSF and PS accumulation in the repair patch.

## Discussion

Lesions in the sarcolemma rapidly lead to formation of repair patches, preventing break-down of the sarcolemmal barrier and cell death. This repair patch has to be removed for restoration of the normal lipid bilayer structure of the sarcolemma. Several mechanisms have been previously suggested for patch removal, including membrane shedding, autophagy and endocytosis[11,34,35]. We show here that macrophages are key players in membrane repair by phagocytosing the repair patch. First, macrophages associate with and phagocytose material from the membrane patch. Second, knock-down of macrophages abolishes repair patch removal. Third, the repair patch rapidly and selectively accumulates PS a known 'eat-me' signal for macrophages[36]. PS is presented at the extracellular side of the repair patch, and thus is accessible to macrophages passing by. Moreover, phagocytosis is delayed when PS is masked.

Repair patch removal by macrophages can occur within minutes in some instances, whereas the clog persisted for hours in other cases. Thus, the repair patch does not seem to produce a long-range attractive signal pointing at the importance of local scanning by macrophages. Whereas small lesions were efficiently repaired, a characteristic sequence of events occurred for larger lesions: formation of a large repair patch, eventually sliding of the repair patch, secondary influx of $Ca^{2+}$, contraction of the muscle cell, precipitation of AnxAs onto internal membranes, cell death and removal of dead cells by macrophages. Surprisingly, neutrophils were not involved in phagocytosis of repair patches or dead cells. We observed in all cases, however, recruitment of neutrophils to muscle injury by stabbing with a glass needle or tailfin cuts. This suggests that inflammatory signals may play additional roles. The overall much higher inflammatory status may explain the difference to myopathic human skeletal muscle, where neutrophils are heavily engaged[37].

Among the lipids tested, only PS and cholesterol accumulated rapidly at lesions, whereas enrichment of general membrane markers was not observed. Our data underscore lipid sorting as mechanism for early repair patch assembly and suggest a key involvement of Dysf in PS-recruitment. Myofibers lacking Dysf failed to accumulate PS. Single-molecule tracking in Dysf morphant and control embryos showed Dysf-dependent, directed movement of PS to the site of lesion. Expression of Dysf fragments in HeLa cells and Dysf-depleted embryos causes PS enrichment. Hence, Dysf sorts PS to the repair patch. It remains to be uncovered whether cholesterol sorting employs the same mechanism.

We noted other lipids, even though not as fast as PS and cholesterol, to accumulate gradually over time. For example, PI(3,4)P$_2$ starts to build up in a delayed manner 4–5 min post lesioning (Supplementary Fig. 7e). This likely reflects membrane flowing in from the edges and providing lipid material to eventually seal the sarcolemmal hole. In this context, systemic exocytosis may be involved expanding the membrane surface of a cell in response to membrane lesions[2].

Binding of PS is mediated by the C2A-domains of Dysf (ref. 38). Our data suggest that the arginine-rich motif identified here plays an important additional role in PS transport. Arginine-rich peptides are known to bind PS (ref. 31). Thus, PS may thus directly interact with the Dysf arginine-rich motif.

Mutation of the arginine-rich motif affects localization of Dysf reporters. They appear to be located in the cytoplasm of uninjured myofibers. This suggests that Dysf has to be localized correctly to the sarcolemma to allow PS accumulation. Our single-molecule tracking data reveal that Dysf moves in the plane of the sarcolemma or immediately adjacent and parallel to it. A potential source of Dysf and PS might be caveolae which unfold on membrane stress and thus rapidly increase membrane

surface[39]. We did, however, not observe enrichment of general membrane markers at the lesion excluding overall flux of membrane material towards the lesion as mechanism of PS transport. The mechanism how Dysf and PS are transported towards the lesion remains elusive and needs further investigation. Another key question is how PS is incorporated in the repair patch so that it is exposed to the extracellular space. High local $Ca^{2+}$ concentrations may inactivate flippases at the lesion as previously proposed in a different context[40].

Zebrafish and human DYSF both contain arginine-rich motifs N-terminally of the TM domain. hWRRFR-TM-C accumulated at the membrane patch and rescued PS accumulation in injured zebrafish myofibers, suggesting that these particular functions of Dysf have been conserved during evolution. This claim is supported by complementary approaches expressing the zebrafish Dysf reporter in HeLa cells; zfWRRFK-TM-C accumulated and mediated enrichment of PS at the site of lesion in human cells.

LGMD2B/MM is correlated with a mutation in the arginine-rich motif (R2042C) in human patients[33]. In accordance, modelling this mutation in both zebrafish and human DYSF fragments revealed significantly lower accumulation at membrane lesions. A disease causing effect of this mutation in humans is thus in complete agreement with our data. Taken together, these human genetic data underscore the physiological significance of the mechanism underlying membrane repair that we uncovered.

## Methods

**Zebrafish strains.** The AB$_2$O$_2$ WT line (European Zebrafish Resource Centre EZRC, Karlsruhe) was used for all experiments. The transgenic lines *Tg(mpeg1:GFP)* and *Tg(LysC:dsRED)* were a gift from the Lieschke and the Crosier labs, respectively[14,15]. Zebrafish husbandry[41] and experimental procedures were performed in accordance with German animal protection regulations (Regierungspräsidium Karlsruhe, Germany, AZ35-9185.81/G-137/10).

**Expression plasmids and sensors.** Cloning was carried out following standard procedures (Supplementary Methods). Muscle expression of sensors was driven by the *unc45b* promoter (ref. 42). The lipid sensors *LactC2:GFP*, *LactC2:RFP*, *GFP-2 × PH(PLCδ)*, *AKT-PH:EGFP*, *Lyn-tailed mCherry-SEpHluorin*[20,23,24,26], secreted human *AnnexinV-YFP* reporter[28], *human DYSF-Venus*[43], *pcDNA3-Clover*[44] and *pEGFP::2FYVE-GFP*[25] were described previously. Please refer to Supplementary Table 2 for a summary of the sensors used.

**Real-time imaging of membrane repair.** Plasmids encoding sensors, BODIPY-cholesterol and TopFluor-PS (Avanti Polar Lipids, AL) were injected into the yolk of 1–2 cell embryos[45]. Levels of expression of sensors did not affect the kinetics of membrane repair processes. Sarcolemmal lesions were generated using 3- to 5-day-old embryos, which were immobilized on a microscopy slide using 0.5% low melting point agarose supplemented with 0.02% MESAB. Embryos were imaged with a water dip-in × 63 objective (NA: 0.90; HCX APO water; Leica) and installed at a Leica TCS SP2 confocal microscope and the corresponding Leica LCS software. The observations were performed at room temperature. The sarcolemma was damaged with a two-photon laser set to 860 nm. Sensor accumulation at the membrane lesion was measured by determining the fluorescence intensity at the lesion in at least five independent experiments. Significance was tested with Welch's test followed by Bonferroni correction using MATLAB.

For imaging (16–22 h), multiple embryos were embedded in LMP agarose (0.5%) in a 6 cm petridish and covered with 10 ml 1 × E3-medium (Supplementary Methods) containing 0.02% MESAB and 0.003% phenylthiourea (PTU). Individual damaged cells were imaged sequentially as Z-stacks of 40–90 μm overnight under an upright TCS SP5 confocal microscope (Leica, HCX PL APO × 20/0.70 lambda blue IMM CORR or HCX APO L × 40/0.80 W U-V-I objectives), using the brightfield and fluorescence channels (488 nm/561 nm).

**Knock-down and knock-out.** Morpholinos against *dysf* mRNA[7] and for depleting macrophages[16–18] were used as described (0.8 mM *dysf–MO*, 0.5 mM *pu.1-MO*; 0.5 mM *gcsfr-MO*; 0.6 mM *irf8-MO*). *dysf* and *irf8* mutants were created by CRISPR/Cas9-mediated stop codon cassette insertion[46].

**Imaging of HeLa cells.** HeLa cells were transfected using Lipofectamine 3000 (Thermo Fisher Scientific, Carlsbad, CA). After 24 h culture at 37 °C, cells were washed with phosphate buffered saline and imaged in DMEM containing 10% foetal bovine serum. Cells were imaged at 37 °C on an Andor Revolution XD

spinning disk confocal laser scanning microscope (BFi OPTiLAS, München, Germany) with an OLYMPUS ApoN60 × /1.49 oil immersion objective. The cell membrane was damaged by using the FRAPPA unit to irradiate a region of $6 \times 6$ pixels with 405 nm laser light at $200 \mu W$ on the specimen with a pixel dwell time of $800 \mu s$. Irradiation was repeated 600 times to locally damage the membrane within 18 s. Image sequences were acquired in 10 s intervals, with the first image taken 20 s after membrane damage, followed by analysis with ImageJ.

**Super-resolution localization microscopy.** Zebrafish myofibers were imaged on a custom-built widefield inverted microscope (Axiovert 200, Zeiss, Göttingen, Germany) with single-molecule sensitivity[47], equipped with a C-Apochromat, × 63/1.2 W Corr objective (Zeiss), multiple excitation lasers (405, 473 and 561 nm), an image splitter (Optosplit, Cairn Research Ltd, Faversham, UK) and an EMCCD camera (Ixon Ultra 897, Andor, Belfast, Northern Ireland). Embryos (4–5 dpf) anesthetized with 0.02% MESAB were immobilized on cover glass surfaces in 1% LMP agarose; another cover slip on top held the embryos closely to the bottom surface. Muscle cell membranes were damaged by focusing 405-nm laser light (5 mW on the specimen) for 2–4 s. mEosFP was photoconverted to its red emitting form by 405 nm light and excited at 561 nm. Image stacks were analysed using a-livePALM software[48] (Supplementary Methods).

**Data availability.** All data are provided in the Supplementary Information.

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

## Acknowledgements

We thank N. Borel and the fish house team, M. Rastegar for microscopy and S. Grinstein, S. Vogel, M. Lin, L.C. Cantley, C. Etard, T. van Ham and R. Peterson for plasmids. We thank T. Dickmeis for critically reading the manuscript. US and GUN were funded by HGF, DFG (STR 439/8-1, Ni 291/12-1), EC IP ZF-HEALTH and BMBF-MIE. In addition, this research work is part of the project "Molecular Interaction Engineering: From Nature s Toolbox to Hybrid Technical Systems", which is funded by the German Federal Ministry of Education and Research (BMBF), funding code 031A095 C.

## Author contributions

V.M., L.Z., G.U.N and U.S designed the experiments V.M. and L.Z. performed experiments. V.M., U.S. and G.U.N wrote the manuscript. M.R. performed statistical analysis. C.G. provided transgenic zebrafish lines. M.S. helped to clone the calcium sensor. M.T. and T.B. contributed to experimental design and cloning. U.R. provided plasmids, S.R. reviewed the manuscript.

## Additional information

**Competing financial interests:** The authors declare no conflict of interests.

