## [Peer Review File · Nature Communications]

Transferred manuscripts:

Reviewers' comments:

Reviewer #1 (Remarks to the Author):

In Middel et al.'s manuscript, "Dysferlin-mediated phosphatidylserine sorting engages macrophages in sarcolemma repair," the authors utilize real-time imaging of membrane damage assays in genetically-modified zebrafish to demonstrate that PS and dysferlin quickly move to the sites of damage to form a 'patch,' which is later phagocytosed by macrophages. Further, they demonstrate that an arginine-rich motif in dysferlin is necessary for this PS transport. The authors present a clear and well-supported argument for their hypothesis, which will be of interest to cell biologists, muscle physiologists, and muscular dystrophy researchers. However, I do have several major concerns:

1. Many of these experiments were done through expression of proteins in a WT zebrafish line, meaning that the proteins are overexpressed. The authors should give an idea of how highly the proteins (AnxA2-mO and Dysf-mO) are expressed above endogenous levels. Also, is there any evidence from them or others to suggest that overexpression of these proteins is sufficient to alter the wound repair capacity of muscle?
2. Figure S1 shows 'precipitation of AnxA2a-mO to internal membranes, a hallmark of cell death.' What accounts for this sudden rise in AnxA2 signal?
3. I am most familiar with macrophage phagocytosis of apoptotic cells. A major difference in the phenomenon described in the current manuscript is that macrophages are phagocytosing just a small patch of membrane and leaving the remainder of the (healthy) cell intact. Is this correct? Are there other systems where macrophages have been shown to perform similar tasks?
4. The text referring to figure 1 describes macrophage 'recruitment.' However, the figure shows injury of a myofiber directly to a site adjacent to a macrophage. Did the authors always choose injury sites that are next to macrophages? How far will macrophages travel to reach a site of injury?
5. The bar graphs in figures S2G and 1D need clarification. Legend states 'repair patch removal in control (blue) and macrophage-depleted embryos (red).' This is incorrect. These two states are depicted on the left (control) and right (macrophage-depleted) bars. The colors indicate the percentage of fibers with patch removed (blue) or patch persisted (red).
6. Figure 2M shows increased scanning time in secA5-YFP conditions. Does secA5 diffuse off after 400min?

7. In Dysf-KD embryos, are there any structural defects or indications that this is replicating LGMD2B disease?
8. Figure 4 explores the lateral movement of sarcolemma after wounding. Have the authors explored the contribution of internal lipid donors (vesicles or organelles)? Or if T-tubule membrane can contribute to the patch? Can PALM reveal this?
9. What is the purpose of repair patch removal? Did the authors follow any fibers where the patch was not removed - did these fibers eventually die? What occurred in macrophage-depleted embryos?

Reviewer #2 (Remarks to the Author):

When the sarcolemma are damaged, they are quickly sealed by forming patches. In this manuscript, Middel et al. show that phosphatidylserine (PS) is rapidly transported from the adjacent sarcolemma to the repair patch in a dysferlin-dependent manner, and the patches are removed by macrophages. This is an interesting proposal. However, the manuscript contains several unclear points. My detailed comments are as follows.

1. Figure 1b: GFP should be specific to macrophages. Yet, a cell in right (probably myofiber cell) is GFP-positive. Why? The strong fluorescence of mOrange is detected in the gate for GFP? Please explain.
2. Figure 1d: They show that the morphants in which triple morpholinos against *gcsfr*, *pu.1*, and *irf8* were injected lost the ability to remove the repair patch. Please explain why three genes should be knocked-down. Knocking down *Pu.1* gene alone is not sufficient? Why *GCSFR* (G-CSF receptor) gene should be knocked-down, if neutrophils are not involved in removing patches?
3. Line 104: They say "Presentation of phosphatidylserine (PS) to the extracellular space". Do the authors want to say that "presentation (or exposure) of PS on the cell surface"? Or, PS-carrying repair patches are released to the extracellular space, as reported by Reddy et al (reference 11)?
4. Paragraph starting from Line 103: The authors claim that since PS quickly accumulates at repair patch, PS is the key component. The quick accumulation itself does not mean that it is the key component. I understand Annexin A2 binds phospholipids. What does AnxA2 recognize here? It does not recognize PS?
5. Paragraph starting from Line 125: PS is localized at the inner leaflet of plasma membrane. Thus Lact-GFP intracellularly recognizes PS. The ecA5-YFP also recognizes PS on repair patch, indicating that PS is exposed at the patch. What kinds of mechanisms do the authors propose? At first, the PS-rich membranes in which PS is localized at inner leaflet recruited to the patch, and then scramble to the external leaflets? What causes the PS-exposure, or externalization? I imagine that if the patch is a part of (or connected to) a myofiber, PS will quickly return to the inner leaflet. The authors should show that the patch is connected or disconnected from the main body of a myofiber.

6. Figure 3: The authors showed that zfWRRFK-TM, a truncated form of Dysferlin, supports the accumulation of PS at repair patch. This is surprising. The authors claim that this molecule selectively transport PS to the repair patch. But, I can not find the data supporting this statement. Other phospholipids are not accumulated at patch with this truncated form of dysferlin? If it is specific, please show with Kd (Dissociation constant) that WRRFK, but not WAAFK specifically binds PS, but not other phospholipids. What kinds of mechanism do the authors propose for the specificity? If Dysferlin binds PS to transport it to repair patch, why Lact-GFP does not compete or inhibit the Dysferlin-mediated PS-transport.

7. The zfWRRFK-TM-C dependent PS accumulation to the repair patch in HeLa cells is interesting. I want to see the result with intact Dysferlin, too. Actually, in line 300, the authors say that "Expression of Dysferlin in HeLa cells causes the PS enrichment." "Dysferlin" here means the intact Dysferlin? If so, the data should be presented.

8. Line 324: The authors confess that "the molecular mechanism how Dysferlin and PS are transported towards the lesion remain elusive". I, too, think that it is too much at this stage to ask the authors to show the molecular mechanism. But, please discuss more. For example, what trigger this movement of PS? Ca^{2+} ? What causes the externalization of PS to the surface of the patch?

Reviewer #3 (Remarks to the Author):

The article by Middel et al describes dynamics of sarcolemmal repair process following muscle injury. This is a nice study that, however, suffers from grammatical issues, lack of sufficient background details of experiments and wrong use of gene/protein symbol nomenclature at many places.

Major Points:

1. Background is very vague: Authors have used words like " part of multimeric complexes", "various vesicles" without giving any information on these. I understand the space limitation but background should be able to provide relevant details leading to the submitted research that non-dysferlin muscle and or zebrafish researcher can easily understand.

2. In line 54: authors point that none of the vesicles that were described in previous studies contributed to membrane repair in zebrafish myofibers. Could these differences be due to species related differences then scientific discrepancy. Authors need to explain that well.

3. Line 114: Please define what does CAAX-mcherry labels? Similarly which membrane does Lyn-tailed mCherry labels? Details lacking on these lines prevents the complete understanding of this experiment. As reporter lines are crucial for this study, it will be highly beneficial to describe all reporter lines in a table.

4. A single morpholino by itself can result in non-specific effects. The use of three morpholinos are a bit unusual for knockdown studies. Please provide the concentrations of these morpholinos and controls employed to rule out non-specific effects.
5. Results from Figure 2 regarding PS are a bit confusing. LactC2-GFP appears at the site of injury within 30 seconds (Figure 2a). Similarly, macrophages were shown to appear much earlier (figure 1) whereas SecA5-YFP appears at 282 seconds. What is the rationale for the concluding that SecA5-YFP prevents phagocytosis? Further what is the significance of cholesterol at rapir patch considering it shows similar rates of accumulation as of PS?
6. Authors describe the use of dysferlin mutant. Has this mutant been published? Does this mutant show similar phenotypes as of morphant fish and or human disease? Why did authors chose to work with morpholino while they already have a genetically clean mutant model available?
7. Authors don't describe why 5-AA motif is essential for PS accumulation.
8. Quality of some of the images is poor. Figure 3, bottom left panel, G', Figure 1 , Panel b.'
9. Authors have used HeLa cells to confirm the conservation between zebrafish and human studies. It will be highly relevant to show these in muscle cells, primary myoblasts or differentiated cells.
10. In this study, authors show that macrophages accumulate at the site of injury something that has already been shown by other studies (Roche et al, 2015, Nagaraju et al., 2008).

Minor points:

1. Line 40: two periods after integrity
2. Gene symbol: DYSF for humans (please correct in line 40)
3. Line 45: What kind of multimeric complexes? Are they dependent or independent of Dysferlin?
4. Line 48: Knockdown is typically associated with gene and hence gene symbols should be used (italicized) or something like Down-regulation of *Dysf* or *AnxA6* translation. Similarly, *Anxa2a-mO* is non-italicized but *Dys-mO* is (lines 107-110).
5. Line 97: Please define observation time
6. Reporter lines are described with italicized symbols in text but with non-italicized symbols in figure legends. These should be uniform (italicized)

Response to reviewers' comments:

Reviewer #1 (Remarks to the Author):

In Middel et al.'s manuscript, "Dysferlin-mediated phosphatidylserine sorting engages macrophages in sarcolemma repair," the authors utilize real-time imaging of membrane damage assays in genetically-modified zebrafish to demonstrate that PS and dysferlin quickly move to the sites of damage to form a 'patch,' which is later phagocytosed by macrophages. Further, they demonstrate that an arginine-rich motif in dysferlin is necessary for this PS transport. The authors present a clear and well-supported argument for their hypothesis, which will be of interest to cell biologists, muscle physiologists, and muscular dystrophy researchers. However, I do have several major concerns:

1. Many of these experiments were done through expression of proteins in a WT zebrafish line, meaning that the proteins are overexpressed. The authors should give an idea of how highly the proteins (AnxA2-mO and Dysf-mO) are expressed above endogenous levels. Also, is there any evidence from them or others to suggest that overexpression of these proteins is sufficient to alter the wound repair capacity of muscle?

The reviewer has raised a valid point which we have addressed in a number of ways. First, our desire was not to alter the wound repair capacity of muscle by expression of the sensors but monitor and measure the repair processes as undisturbed as possible. The only way we could take into account the endogenous levels of the involved proteins is by employing a suite of antibodies that work on whole mounts or tissue sections. Unfortunately, these antibodies do not yet exist for the zebrafish proteins, and this approach would prevent kinetic analysis as it would entail fixation of the material. We present additional lines of evidence which suggest that overexpression does not significantly disturb the repair process.

We do not observe strong variations in the kinetics of patch formation measured by accumulation of the PS sensor LactC2:GFP (Fig. A),

Fig. A Comparison of LactC2:GFP accumulation over time as single (green) or co-injections with AnxA2a-mO (red), Dysf-mO (purple) or CAAX-mO (blue). There is no difference in LactC2:GFP accumulation in the different co-injections, indicating that overexpression of AnxA2a-mO, Dysf-mO or CAAX-mO together with LactC2:GFP does not influence the accumulation of PS in the repair patch.

despite the fact that the injured muscle fibres do express the sensors at different levels (Fig. B).

Fig. B Variation of overexpression of LactC2:GFP in undamaged myofibers. All mean values for single LactC2:GFP expressing cells were quantified using ImageJ by drawing a rectangular area across each overexpressing cell. The mean value of all LactC2:GFP (n=16) expressing cells was set to 100%, followed by calculating the relative fluorescence for each overexpressing cell. Although the variation in expression is high, the accumulation kinetics of the same experiments are similar (see one slide before). This indicates that mere overexpression, be it single or co-injections, does not influence the accumulation of PS.

This variation in expression is not correlated with altered kinetics of patch formation (Fig. A). When we express different sensor combinations (AnxA2a-mO or Dysf-mO with the PS sensor or the PS sensor alone), we observe similar kinetics, showing that expression of any one of these sensors does not affect repair patch formation. Moreover, in each case of “overexpression”, we observed accumulation of macrophages and phagocytosis of repair patches.

Moreover, loss of Dysf (by knock-down or mutation) led to loss of PS accumulation. This can be rescued to similar levels and with similar kinetics by forced expression of the short Dysf fragments in comparison to the situation when the PS sensor was expressed alone and we had to rely on the action of endogenous Dysf (Fig. 3i, extended Data Fig. 4h-j in the manuscript).

Thus, the repair system tolerates expression of the sensor proteins. This is consistent with other studies in the field of membrane repair (e.g., Lostal et al. 2012; Liu et al. 2015b; Glover et al. 2010). Taken together, we assert that it is unlikely that the levels of expressed sensor proteins play a big role in the responses at all. We included a statement in the materials and methods section to indicate this: “Levels of expression of sensors did not affect the kinetics of membrane repair processes“(page 14).

2. Figure S1 shows 'precipitation of AnxA2a-mO to internal membranes, a hallmark of cell death.' What accounts for this sudden rise in AnxA2 signal?

It is in fact not a sudden rise in AnxaA2a-mO but rather the sudden precipitation of the cytoplasmic protein onto internal membranes. This redistribution from a dispersed cytoplasmic location appears like a rise in concentration. It is well established that Ca^{2+} ions trigger this association of annexins with lipid bilayers. As documented in Figure S1E to E", myofibers with large lesions will eventually lead to a massive influx of Ca^{2+} ions from the site of lesion which, in addition, will trigger the release of Ca^{2+} ions from internal stores, leading to the systemic precipitation of AnxA2a-mO on all internal membranes followed by death and phagocytosis of the damaged cells. This particular cell fate was typically observed when cells were damaged with lesions larger than 4 μm .

3. I am most familiar with macrophage phagocytosis of apoptotic cells. A major difference in the phenomenon described in the current manuscript is that macrophages are phagocytosing just a small patch of membrane and leaving the remainder of the (healthy) cell intact. Is this correct? Are there other systems where macrophages have been shown to perform similar tasks?

Yes, indeed, the reviewer is correct. In our case, macrophages remove the repair patch and leave the cell intact. In fact, our data show that full restoration of the cell membrane after wounding, i.e., removal of the repair patch, requires this interaction with macrophages. To the best of our knowledge, there is no other system described in which macrophages embark on a similar function. In a very recent publication in *Cell*, macrophages were shown in a different context to fuse ends of laser cut blood vessels by contraction force (Liu et al., 2016).

4. The text referring to figure 1 describes macrophage 'recruitment.' However, the figure shows injury of a myofiber directly to a site adjacent to a macrophage. Did the authors always choose injury sites that are next to macrophages? How far will macrophages travel to reach a site of injury?

Thanks for pointing this out. No, we did not at all choose fibers close to a macrophage. This was merely an incident where a macrophage happened to be close to a labelled cell. Macrophages distribute randomly in somitic muscle. We show an example in Fig. D. The first macrophages appeared in the field of view at 240 min after lesioning; it arrived at 1080 min at the site of lesion and moved away again after phagocytosis of the patch at 1110 min.

Fig. D Double transgenic animals with macrophages (green, *mpeg1:GFP*) and neutrophils (red, *LsyC:dsRED*) were injected with *anxa2a-mO* to label individual muscle fibers in the somitic musculature of zebrafish embryos. A lesion was inflicted in a single myofiber, and the approach of macrophages and neutrophils was monitored over 1110 min. 540 min after lesioning, a macrophage closed in and eventually settled on the lesion. By 1110 min the macrophage had phagocytosed the patch and departed.

5. The bar graphs in figures S2G and 1D need clarification. Legend states 'repair patch removal in control (blue) and macrophage-depleted embryos (red).' This is incorrect. These two states are depicted on the left (control) and right (macrophage-depleted) bars. The colors indicate the percentage of fibers with patch removed (blue) or patch persisted (red).

We agree with the reviewer. This is an error and a very unusual presentation of data of two outcomes only. We changed this presentation and now give the percentage of injured myofibers without repair patch. The remaining myofibers retained their patch (shown in red previously) and are not explicitly considered in the new version of the graph. As a consequence, there are 0% injured myofibers without repair patch in the knock-down. All injured cells retained the lesion patch in the absence of macrophages. We replaced the graphs in the manuscript.

6. Figure 2M shows increased scanning time in *secA5-YFP* conditions. Does *secA5* diffuse off after 400min?

We have no evidence of diffusion of the SecA5 PS sensor with time. The measurements shown in Figure 2I suggest that *secA5* still increases at the repair patch up to 270 s post-lesioning and persists until at least 1140 min at high levels (Fig. E). Clearly, a macrophage can recognise a masked patch as it would otherwise not settle in the vicinity of the patch. Such dominant negative approaches are rarely as efficient as knockouts and we may just witness leakiness of the system. We can also not exclude that other signals triggering phagocytosis are involved. It may well be that *secA5-YFP* cannot mask PS efficiently over time due to dissociation and diffusion as suspected by the reviewer. However, we did not see significant loss of *secA5-YFP* until at least 1140 min (Fig. E). Irrespectively, our data clearly demonstrate interference with

phagytosis (i.e., a delay), in line with the other evidence that PS and macrophages play a crucial role in repair patch processing.

We never observed dissociation of the patch from the injured myofiber over the entire observation times (Fig. E) including those experiments where macrophages were removed.

Fig. E secA5-YFP at the lesion patch has not diffused away after 1140 min and the repair patch remains in association with the injured myofiber. Please note that there is bleed-through from the YFP into the GFP channel. Therefore, the fiber artificially appears to be GFP positive.

7. In Dysf-KD embryos, are there any structural defects or indications that this is replicating LGMD2B disease?

Yes, we observe failure to transport PS efficiently to the lesion patch in the *dysf* morphants as well the *dysf* mutant, in line with the defective membrane repair ability in LGMD2B patients. In addition, we observed an effect on birefringence of the somitic musculature in the morphants, which is an indication of impaired myofibril organisation. This impaired myofibrillar organisation was, however, not observed in the mutant. The predominant purpose was to use the mutant to show that the effects that we see by knock-down of *dysf* expression in repair patch formation can be reproduced by a gene knock-out. We developed the genetic *dysf* knock-out only recently and, therefore, an in-depth analysis of the zebrafish mutant is not available yet. A detailed analysis, especially also of the adult musculature, has not been carried out yet and would be a considerable effort far beyond the scope of this manuscript.

8. Figure 4 explores the lateral movement of sarcolemma after wounding. Have the authors explored the contribution of internal lipid donors (vesicles or organelles)? Or if T-tubule membrane can contribute to the patch? Can PALM reveal this?

We had systematically investigated the following vesicle types by monitoring their accumulation at the repair patch: Laptm4a, Lamp1, Lamp2, Rab1a, Rab5a, Rab6a, Rab7 and Rab27a containing vesicles. None of these vesicles happened to be integrated in a significant way into the repair patch in our system (Roostalu and Strähle, Dev Cell, 2012). We thus believe that vesicular transport to the lesion plays a minor role if any.

We agree that T-tubules could be a source of repair membrane material. Also, the junctions to the sarcolemma are frequent locations of membrane tears. Moreover we found Dysf protein localized in the T-tubule membrane. Unfortunately, our initial single molecule tracking experiments contain too few tracks originating in the T-tubules and even after repeating the experiment, we cannot reach a strong conclusion (Fig. F). We cannot rule out a contribution by

Fig. F In this experiment, the myofiber was first damaged by 405 nm laser irradiation (red arrow). However, 405 nm laser exposure also leads to strong local photoactivation (green-to-red photoconversion) of mEosFP*thermo*, which precludes single molecule localization-based imaging in this region. Therefore, after membrane damage, strong 561 nm laser irradiation was applied to bleach these activated mEosFP*thermo* fluorophores in the red channel for about 1 min so as to be able to start recording individual trajectories. Consequently, a large fraction of activated mEosFP*thermo* fluorophores in the damaged region was bleached, and in the ensuing single molecule tracking experiments, there are only a few trajectories from the surrounding T-tubule area (blue square), whereas the measured membrane area (red square) further away from the lesion is densely labeled. With only a small number of trajectories in the T-tubule region, it is not possible to quantify single molecule motion in this region.

T-tubule membrane to the repair patch. However, for technical reasons, we cannot unequivocally answer the question.

9. What is the purpose of repair patch removal? Did the authors follow any fibers where the patch was not removed - did these fibers eventually die? What occurred in macrophage-depleted embryos?

This is a very difficult question to address as we are limited by our experimental set-up. Damaged cells were followed until 16-20 h. Until this stage, they retained the patch and were healthy. Morpholino effects on macrophages will wear off with time and macrophages will reappear, preventing a long term study. The same holds true for the *irf8* knockout line (Shiau et al. PlosOne 2015). For these experiments with up to 20 h of observation, the embryos needed to be immobilized. Otherwise, we would not have been able to image them at this high resolution. Thus, the fibers are not exposed to their normal mechanical stress. With current technology, we do not see a way how to address this question. As most of the patches are removed after 20 h in wildtype embryos, we assume that the removal by macrophages is necessary to restore the normal lipid bilayer of the cell membrane again.

Reviewer #2 (Remarks to the Author):

When the sarcolemma are damaged, they are quickly sealed by forming patches. In this manuscript, Middel et al. show that phosphatidylserine (PS) is rapidly transported from the adjacent sarcolemma to the repair patch in a dysferlin-dependent manner, and the patches are removed by macrophages. This is an interesting proposal. However, the manuscript contains several unclear points. My detailed comments are as follows.

1. Figure 1b: GFP should be specific to macrophages. Yet, a cell in right (probably myofiber cell) is GFP-positive. Why? The strong fluorescence of mOrange is detected in the gate for GFP? Please explain.

Yes, as the reviewer points out, this is due to bleed-through of the Anxa2a-mOrange signal into the GFP channel. Since the cells have such a different shape, we do not consider this to be a problem for the interpretation of our data. We introduced a comment into the legend of Figure 1 to explain this: “Note the bleed-through of AnxaA2a-mO in GFP channel (b to b)”. “

2. Figure 1d: They show that the morphants in which triple morpholinos against gcsfr, pu.1, and irf8 were injected lost the ability to remove the repair patch. Please explain why three genes should be knocked-down. Knocking down Pu.1 gene alone is not sufficient? Why GCSFR (G-CSF receptor) gene should be knocked-down, if neutrophils are not involved in removing patches?

We initially assumed that neutrophils and macrophages will remove the patch. So we chose this approach from the literature. By looking at transgenics marking neutrophils, we found that they are, unlike macrophages, not totally removed, and they also did not interact with the lesion patch; so we turned our attention to macrophages. We systematically tested each morpholino alone and quantified the macrophages after morpholino knock-down. In controls (n=44) and pu.1 (n=11), GCSFR (n=17) and irf8 (n=29) single knock-downs macrophages were still present in large numbers. We decided to go for a triple knock-down (n=31 embryos tested). Only in this case, macrophages were strongly reduced or totally eliminated (15 out of 31 embryos showed a few remaining macrophages). In the meantime, we have obtained results from an *irf8* Crispr/Cas9 knock-down that also eliminates macrophages (Fig. K, see response to comment 4 of reviewer 3). Thus, the fact that we needed a triple knock-down most likely indicates that single morpholinos partially lose their effect already at 3 days post-fertilization; only the combined action of all three knock-downs kept the macrophage count sufficiently down.

3. Line 104: They say "Presentation of phosphatidylserine (PS) to the

extracellular space". Do the authors want to say that "presentation (or exposure) of PS on the cell surface"? Or, PS-carrying repair patches are released to the extracellular space, as reported by Reddy et al (reference 11)?

Excuse us, please. None of us is a native English speaker. In our humble understanding of the English language, actually both expressions would be correct. You need to be exposed to be presented. But I guess one would call it exposed if it were, as we said, innate like an extracellular space. For macrophages, "presentation" may be better. However, as we use it in combination with "extracellular space", we accept the reviewer's point and changed the word "presentation" to "exposure".

With respect to release of repair patches into the extracellular space, we did not observe this at all, even in experiments up to 20 h after wounding. Please see also Fig. E of this rebuttal letter with additional supporting information. The difference between our study and studies suggesting repair patch shedding is that we studied repair patch removal *in vivo*. Reddy et al used, for example, tissue culture of fibroblasts not even muscle cells. We did not see release of the repair patch into the extracellular space in our *in vivo* system with an intact tissue context.

4. Paragraph starting from Line 103: The authors claim that since PS quickly accumulates at repair patch, PS is the key component. The quick accumulation itself does not mean that it is the key component. I understand Annexin A2 binds phospholipids. What does AnxA2 recognize here? It does not recognize PS?

We may have been imprecise here. What we wanted to say is that PS is "a" key component (i.e., one of several but not the only one) of the early lesion patch. As we have pointed out in a previous publication (Roostalu and Strähle, Dev. Cell.), Dysf and annexins arrive at the lesion patch in a coordinated time sequence. The time of arrival thus matters with Dysf and AnxA6 as early components followed by AnxA2a and others. It is important that PS is a component of the early repair patch, as this will form the structure that is exposed to the exterior. We state now at the end of the paragraph: "Clearly, PS is one key component of the early repair patch."

In principle, annexins are capable of binding phospholipids and form multimeric crystal-like complexes in a calcium dependent manner (Lennon et al. 2003; Gerke et al. 2005). In particular, binding of annexin A2 to the plasma membrane depends on phosphatidylinositol-4,5-biphosphate [PI(4,5)P₂] and calcium (Rescher and Gerke 2004; Rescher et al. 2004). Annexins may recognize PS, or remnants of the still existing plasma membrane surrounding the hole giving stability and tightness. There are also functions of annexin2 in mammals that are phospholipid-independent and rely on protein-protein interaction (Rescher and

Gerke, 2004).

5. Paragraph starting from Line 125: PS is localized at the inner leaflet of plasma membrane. Thus, Lact-GFP intracellularly recognizes PS. The ecA5-YFP also recognizes PS on repair patch, indicating that PS is exposed at the patch. What kinds of mechanisms do the authors propose? At first, the PS-rich membranes in which PS is localized at inner leaflet recruited to the patch, and then scramble to the external leaflets? What causes the PS-exposure, or externalization? I imagine that if the patch is a part of (or connected to) a myofiber, PS will quickly return to the inner leaflet. The authors should show that the patch is connected or disconnected from the main body of a myofiber.

As a prelude to our answer:

A membrane tear is approximately similar to the impact of a torpedo in a ship's hull. The result is a hole and ruptured edges. Ca^{2+} ions will flow in and activate aberrantly the Ca^{2+} -dependent proteases of the caspase family; many other calcium dependent processes are at least locally disturbed. The reducing internal milieu of the cell is locally changed to an oxidative one and, moreover, the ion gradients across the membrane (Na^+/K^+ in addition to Ca^{2+} , etc) will break down completely. Molecules essential to drive the cellular processes will diffuse to the outside etc..

Our data suggest that PS is transported from the intact membrane flanking the lesion towards the lesion. How could PS remain exposed to the outside of the cell? Activated caspases could inactivate flippases around the edge of the lesion as proposed (Segawa et al. 2014). Alternatively, vesicles may be formed at the edge of the plasma membrane, or Dysf may aggregate in such a way in the lesion patch that PS is extracted from the lipid bilayer. However, this is, at this point in time, pure speculation. The mechanism how PS gets exposed is totally unclear and requires an entire new study. Here we provide the first overall evidence (1) that PS accumulates in the lesion patch, (2) that it can be seen at the outside and is sensed by macrophages, and (3) that macrophages remove the lesion patch. (4) Transport requires Dysf, and (5) relevant sequences are conserved in human Dysf and are linked to an allele causing LGMD.

All patches that we monitored in 25 time lapse movies for up to 20 h remained in contact with the injured myofiber. See also Fig. E.

6. Figure 3: The authors showed that zfWRRFK-TM, a truncated form of Dysferlin, supports the accumulation of PS at repair patch. This is surprising. The authors claim that this molecule selectively transport PS to the repair patch. But, I can not find the data supporting this statement. Other phospholipids are not accumulated at patch with this truncated form of dysferlin? If it is specific, please show with Kd (Dissociation constant) that WRRFK, but not WAAFk specifically binds PS,

but not other phospholipids. What kinds of mechanism do the authors propose for the specificity? If Dysferlin binds PS to transport it to repair patch, why Lact-GFP does not compete or inhibit the Dysferlin-mediated PS-transport.

We have refrained from stating that we present evidence suggesting direct transport, as our current data do not support this claim. We thus totally agree with the reviewer. We have considered a number of approaches to measure the direct interaction of zfWRRFK-TM with PS. Please note that both the WRRFK motif and the TM domain are highly conserved between zf and human dysf (as shown in Fig. 6a). We thus believe that, in addition to the interaction of the WRRFK motif with the head group of PS on the surface of the inner leaflet of the bilayer, there is also the interaction of the hydrophobic TM domain with the rest of the PS buried in the lipid layer. Thus, there are not only interactions of the PS head group with the WRRFK motif but also hydrophobic interactions of the DYSF TM with the PS lipid tails. It is an entirely new and technically challenging project to synthesize functional zfWRRFK-TM and to reconstitute such a system in vitro to measure the requested Kds.

To verify our conclusions that zfWRRFK-TM co-migrates with PS, we employed dual-color line scanning FCS (2c lsFCS), a powerful method to measure diffusion of molecules in the cell membrane (Dörlich et al., 2015). In this approach, the focused laser spot of a confocal microscope is repeatedly scanned horizontally with respect to the cover glass through a vertical part of the plasma membrane, while intensity time traces are recorded. 2c lsFCS is a cross-correlation method to reveal co-migration of two differently labeled molecules on the cell membrane. The fact that there is a measurable cross-correlation amplitude clearly indicates co-migration of zfWRRFK-TM and the PS marker LactC2:GFP in the HeLa cell membrane. The experiment cannot reveal, however, if zfWRRFK-TM and PS form a well-defined complex or are part of a larger (but not macroscopic!) patch.

Fig. G Preliminary 2c IsFCS experiments on HeLa cells co-transfected with zfWRRFK-TM-C:mCherry and LactC2:GFP to examine if they co-migrate; the resulting data and fits are plotted. The red and green data and curves represent the autocorrelation functions of zfWRRFK-TM-C:mCherry and LactC2:GFP, respectively, and the blue curve represents the cross-correlation of the two channels. The fact that there is a measurable cross-correlation amplitude clearly indicates co-migration of zfWRRFK-TM and the PS marker LactC2:GFP in the HeLa cell membrane. The experiment cannot reveal, however, if zfWRRFK-TM and PS form a well-defined complex or are part of a larger (but not macroscopic!) patch.

Our additional lines of evidence supporting an involvement of the WRRFK motif in the transport of PS are:

- a) Single molecule tracks are directed towards the lesion in wildtype embryos but not in embryos lacking *dysf* (Fig. 4g). zfWAAFk-TM-C shows motion towards the membrane lesion relative to the membrane marker CAAX (Fig. 4f).
- b) In zebrafish embryos where endogenous *dysf* was removed, zfWRRFK-TM-C but not zfWAAFk-TM-C rescues PS accumulation (Fig. 3i, Extended Data Fig. 4h, i, j).
- c) We observed the same dependence of PS accumulation at the lesion in HeLa cells expressing the WRRFK wildtype but not the WAAFk mutant. (Fig. 5f-g)

We carefully checked our statements in the manuscript, and we changed the title of the paragraph describing the physical measurements of molecule movements to: “Dysf mediates PS relocation from the sarcolemma towards the lesion”. The title of the next paragraph was changed to: “Zebrafish Dysf fragment mediates PS accumulation at membrane lesions in human cells”.

The reviewer is absolutely correct in his statement:

“The authors claim that this molecule selectively transport PS to the repair patch. But, I can not find the data supporting this statement. Other phospholipids are not accumulated at patch with this truncated form of dysferlin?”

With the exception of cholesterol we did not note other lipids (at least among the ones that we checked) to be transported to the early lesion patch. Clearly this claim is not precise, as we cannot rule out that cholesterol and other lipids that we have not checked may be transported. For this reason, we have formulated our conclusions very carefully throughout the manuscript. For example in the discussion, we state: “It remains to be uncovered whether cholesterol sorting employs the same mechanism.”

7. The zfWRRFK-TM-C dependent PS accumulation to the repair patch in HeLa cells is interesting. I want to see the result with intact Dysrlin, too. Actually, in line 300, the authors say that “Expression of Dysferlin in HeLa cells causes the PS enrichment.” “Dysferlin” here means the intact Dysferlin? If so, the data should be presented.

We apologise for this confusion. In line 300, we refer to experiments where zfWRRFK-TM-C was used. We corrected this error. We used human full length Dysf, and as was shown, the cDNA does not include alternative Exon40a, which is needed to cut Dysf into “miniDysf”, which then accumulates in the repair patch (Lek et al., 2013). Therefore our approaches using full length Dysf in HeLa, C2C12 and zebrafish larvae led to expressing cells/fibers with correct localization in the uninjured state. However, no accumulation at the lesion patch could be observed due to failure of correct processing. Intact mouse Dysf:eGFP was transfected into both HeLa cells (Fig. H) and C2C12 myoblasts (Fig. I).

Fig. H Transfection of full length Dysf:eGFP into HeLa cells. The confocal images show the Dysf:eGFP distribution before and at various time points after lesioning at the site marked by the arrows.

Fig. 1 Transfection of Dysf:eGFP into C2C12 cells. A lesion was inflicted (arrows) and the distribution of GFP fluorescence was monitored afterwards, as shown here at various times

The plasmid was a kind gift from Michele's group (J. McDade and D. Michele, 2014). Both in HeLa cells and C2C12 myoblasts, intact Dysf showed correct localization but no accumulation at the lesion after membrane damage. Human intact Dysf was also tested in both HeLa cells and C2C12 myoblasts and showed similar results. This is because the cDNAs do not contain the alternative Exon40a, which is needed to cleave Dysf to generate 'miniDysf', which then accumulates at the lesion (Lek et al., 2013).

8. Line 324: The authors confess that "the molecular mechanism how Dysferin and PS are transported towards the lesion remain elusive". I, too, think that it is too much at this stage to ask the authors to show the molecular mechanism. But, please discuss more. For example, what trigger this movement of PS? Ca^{2+} ? What causes the externalization of PS to the surface of the patch?

We totally agree with the reviewer. The space limitations of most journals are at times suffocating. We still managed to squeeze in a bit more discussion of the issues raised by the reviewer. The accumulation of zfWRRFK-TM-C in HeLa cells is indeed calcium dependent. We added this information and tried to summarize ideas as to how externalisation could happen. As the reviewer sees from our response to his point 5, there are a lot of possibilities but not enough experimental data (particularly, high resolution imaging) of the processes in the region surrounding the patch. We added to the discussion: "Another key question is how PS is incorporated in the repair patch so that it is exposed to the extracellular space. Ca^{2+} -dependent caspases may locally inactivate flippases at the lesion as previously proposed in a different context (Segawa et al. 2014)."

Reviewer #3 (Remarks to the Author):

The article by Middel et al describes dynamics of sarcolemmal repair process following muscle injury. This is a nice study that, however, suffers from grammatical issues, lack of sufficient background details of experiments and wrong use of gene/protein symbol nomenclature at many places.

We are very grateful that the reviewer has taken the time to go to such a depth of constructive input.

Major Points:

1. Background is very vague: Authors have used words like " part of multimeric complexes", "various vesicles" without giving any information on these. I understand the space limitation but background should be able to provide relevant details leading to the submitted research that non-dysferlin muscle and or zebrafish researcher can easily understand.

We apologise. We added more detail to the introduction.

2. In line 54: authors point that none of the vesicles that were described in previous studies contributed to membrane repair in zebrafish myofibers. Could these differences be due to species related differences then scientific discrepancy. Authors need to explain that well.

Reference to an involvement of vesicles is numerous in the mammalian literature. We earlier conducted an extensive study (Roostalu and Strähle, Dev Cell 2012) to investigate these processes more thoroughly in the zebrafish, with the aim to develop the zebrafish as a model for membrane repair. To our surprise, we could not find any evidence of vesicular involvement in immediate repair patch formation, with the very exception of the occasional inclusion of a nearby vesicle into the repair patch. Please see response to comment 8 of reviewer 1 for the list of vesicular markers tested in our previous study. We have added further information in the introduction. As we have discussed these issues in our previous publication, we hope that the reviewer understands that we would not like to expand in this direction too broadly. It addresses an issue that we cannot solve without addressing it experimentally in the mammalian systems using our real time quantitative kinetic techniques. This is a new story line.

3. Line 114: Please define what does CAAX-mcherry labels? Similarly which membrane does Lyn-tailed mCherry labels? Details lacking on these lines prevents the complete understanding of this experiment. As reporter lines are crucial for this study, it will be highly beneficial to describe all reporter lines in a table.

CAAX is a prenylation motif that, when fused to a mCherry fluorescent reporter, is a very popular marker for membranes. Lyn-tailed is the membrane targeting domain of the Src-family kinase Lyn, which was fused to mCherry to target the FP to the membrane. This domain gets lipidated by N-myristoylation and S-palmitoylation. As such, Lyn is also a marker of membranes in general. We thank the reviewer for the suggestion of summarizing the different reporter lines in a table. We have included this table in the materials and methods section as part of the supplemental material (Supplementary Table 2)

4. A single morpholino by itself can result in non-specific effects. The use of three morpholinos are a bit unusual for knockdown studies. Please provide the concentrations of these morpholinos and controls employed to rule out non-specific effects.

Concentrations of morpholinos directed against pu.1 (final 0.5 mM); gcsfr (final 0.5 mM); irf8 (final 0.6 mM) are as given. We added this to the materials and methods section. Accordingly, the total concentration of morpholinos was 1.6 mM. The control was set to 1.6 mM as well. In order to check whether the triple MO knock-down influences the repair process, the relative fluorescence of AnxA2a-mO in the repair patch was measured and quantified (see below Fig. J). As can be seen from the data, the accumulation is normal and comparable to AnxA2a-mO accumulation shown in Figure 2a. Also, embryos developed normally and muscle birefringence is normal in triple morphants. Please also see our remarks to comment #2 of Reviewer 2.

Fig. J Accumulation of AnxAa2-mO at the repair patch over time in embryos in which macrophages were knocked-down (red) by triple injection of Mo-pu.1 (final 0.5 mM); MO-gcsfr (final 0.5 mM); MO-irf8 (final 0.6 mM) versus control MO injected zebrafish (green). 150% relative fluorescence was reached at ~155 s (control) and ~140 s (KD).

Irrespectively, we anticipated criticism along these lines and established an *irf8* knock-out line by CRISPR/Cas9 technology while the paper was under review. In contrast to the morpholinos, *irf8* single knock-out is sufficient to eliminate macrophages, at least up to the developmental stages that we used to analyse the repair of the membrane lesions. As a positive control, damage to several

myofibers was induced and the removal of dead cells by macrophages was checked over 18 h. While in WT siblings, dead fibers had been removed (n=10 experiments), dead myofibers persisted in *irf8*-KO fish (n=10 experiments) over the whole imaging time of 18 h, as seen for the triple knock-down (p=***). Thus, the requirement of a triple knock-down (see also answer to comment 2 of reviewer 2) is most likely due to the leakiness of the individual morpholinos. We also checked the removal of repair patches in injured myofibers (Fig. K).

Fig. K Crispr/Cas9 knock-out of *Irf8* impairs membrane repair patch removal. In controls, the repair patch was removed in 32% of injured myofibers (n=19) within one hour after injury. In the *irf8*-KO embryos the patch was present in all cases of injured myofibers examined (n=15). Significance was checked with Fishers exact test at $p < 0.05$ ($p=0.023895$ *).

The patches were removed in 32% of injured myofibers in wildtype siblings. In contrast, none of the patches were removed in *irf8* mutant embryos (Fig. K). These results confirm our findings of the triple knock-down by an independent and genetic method and fully support our conclusions.

5. Results from Figure 2 regarding PS are a bit confusing. LactC2-GFP appears at the site of injury within 30 seconds (Figure 2a). Similarly, macrophages were shown to appear much earlier (figure 1) whereas SecA5-YFP appears at 282 seconds. What is the rationale for the concluding that SecA5-YFP prevents phagocytosis? Further what is the significance of cholesterol at rapir patch considering it shows similar rates of accumulation as of PS?

We are puzzled why the reviewer derived this rather confused picture of the sequence of events from our graphs. Indeed, PS accumulates as rapidly as 18 s (as measured by LactC2-GFP accumulation), which matches that of the Dysf fragment. Macrophages appear at the injured myofibers within the following hours. The precise time of engagement appears to depend on the accidental arrival of the macrophage, so that it is close enough to sense the repair patch. In

the particular case at hand, the macrophage interacted with the repair patch for the first time at 240 s. It was close to the myofiber at the time of injury. AnxA2a-mO accumulation had started already roughly 60 s after injury. Like LactC2-GFP, accumulation of SecA5-YFP, the extracellular PS sensor, had commenced to accumulate also already at 18 s (Figure 2I, quantification). Since AnxA5A-YFP is secreted, it binds only to PS if that is available extracellularly. Therefore, we proposed that SecA5A might be capable of masking PS, which in turn should lead to a delay in recognition of repair patches. **This is exactly what we observed.**

Cholesterol is another component of the lipid bilayer that gets enriched in the lesion patch. How it accumulates and for what reason is unclear and part of further investigations. One probable explanation is that cholesterol is known to stabilize biological membranes. As the patch and the surrounding sarcolemma is already destabilized due to damage, it would make sense that cholesterol accumulates. We have retained the data on cholesterol in the manuscript because they indicate that there is at least one other lipid that arrives early. The severe limitations on space in the manuscript prevent a detailed analysis or even discussion of cholesterol transport and function in the lesion patch. They are avenues to explore in future work.

6. Authors describe the use of dysferlin mutant. Has this mutant been published? Does this mutant show similar phenotypes as of morphant fish and or human disease? Why did authors chose to work with morpholino while they already have a genetically clean mutant model available?

We kindly refer the reviewer also to the answer to comment 7 of reviewer 1. The mutant has not yet been published. The mutant recapitulates the defect of the morphant in the repair patch, i.e., loss of PS accumulation. The mutant does not show the effect on muscle birefringence seen previously for the *dysf* morphants. This may be the result of maternal contribution, compensatory or unspecific effects. Currently, we also cannot exclude that this is a reflection of the slightly different strategies used to knock down or knock out *dysf*. The major goal for this study was to reproduce the morphant phenotype with respect to repair patch formation and PS accumulation. Further studies, also in adult zebrafish to assess the effects on muscle physiology and structure, are needed to fully describe the mutant and answer the questions of the reviewer.

7. Authors don't describe why 5-AA motif is essential for PS accumulation.

We kindly refer the reviewer also to the answer to comment 8 of reviewer 2. As stated above, we believe that a comprehensive and convincing analysis of this question is far beyond the scope of this paper. We show several lines of indirect evidence that the 5AA motif is required for PS accumulation at the lesion site. We believe that the 5AA motif contributes to the interaction of Dysf with PS in the membrane. Likely, the conserved TM domain is also involved. However, the details of how this achieved, whether this is a one by one interaction or part

of a larger membrane assembly, as we already suggested earlier (Roostalu and Strähle Dev Cell 2012), we presently do not know. This requires new experiments using complementary approaches and stretches current imaging technology close or beyond its limits. We tried hard to get closer to an answer. By doing so, we realized that this will require several years of extra work.

8. *Quality of some of the images is poor. Figure 3, bottom left panel, G', Figure 1 , Panel b.'*

We checked this. All images have now 300 dpi, as requested by Nature Communications.

9. *Authors have used HeLa cells to confirm the conservation between zebrafish and human studies. It will be highly relevant to show these in muscle cells, primary myoblasts or differentiated cells.*

We have repeated these experiments in the requested cell types. The results in C2C12 myoblast are consistent with those in HeLa cells, as discussed in the manuscript. Upon sole expression of the PS sensor LactC2:GFP, there is no significant enrichment in the lesion patch after 405 nm laser induced membrane damage over the following 120 s (Fig. L).

Fig. L C2C12 myoblasts transfected with *LactC2:GFP* alone. Arrows indicate position of membrane lesion at different time points after wounding.

However, co-expression of LactC2:GFP and zfWRRFK-TM-C:mCherry led to significant accumulation of PS at membrane lesions in the same time interval (Fig. M).

Fig. M Co-expression of LactC2:GFP and zfWRRFK-TM-C:mCherry in C2C12 myoblasts. Site of lesion (arrows) and times after wounding are indicated.

Imaging in the red channel showed clear accumulation of zfWRRFK-TM-C:mCherry (Fig. N), which is strongly colocalized with PS in the green channel.

Fig. N Imaging of panels in Fig. M in the red channel to visualise accumulation of PS at membrane lesion in C2C12 myoblasts. Arrows indicate the lesioning site. Different time points after wounding are shown.

The C2C12 myoblast cells expressing LactC2:GFP were also differentiated into myotubes to carry out the same membrane damage experiment. The results were different from those with myoblasts. PS showed substantial enrichment at the membrane lesion after laser-induced membrane damage, both at the cell bottom (Fig. O) and the side membrane (Fig. P).

Fig. O Accumulation of PS in repair patch in differentiated myotubes is independent of a co-expressed Dysf fragment and relies on endogenous Dysf.

Fig. P Accumulation of PS in the repair patch of differentiated myotubes is independent of a co-expressed Dysf fragment and relies on endogenous Dysf.

These results on C2C12 myoblasts and myotubes are most insightful and underscore the functional role of endogenous Dysf. In the myoblast stage, there is barely any endogenous Dysf expressed (Doherty et al. 2005), and many experiments show that the intact Dysf does not have the ability to accumulate at this stage, which means that endogenous Dysf does not influence the behavior of

PS and zfWRRFK-TM. Consequently, the experimental results with C2C12 myoblast are completely in line with those using HeLa cells. However, differentiated myotubes express a considerable amount of endogenous Dysf, which is known to accumulate at the lesion. This functional, endogenous Dysf can rescue PS accumulation in the way that zfWRRFK-TM does this in HeLa cells and C2C12 myoblast. As a result, the PS sensor LactC2:GFP shows significant enrichment at the membrane lesion even for a single transfection.

10. In this study, authors show that macrophages accumulate at the site of injury something that has already been shown by other studies (Roche et al, 2015, Nagaraju et al., 2008).

Yes, we totally agree. It is a well established fact that macrophages remove dead cells. The major difference between our study and previous studies is that we aim at simulating an isolated membrane tear that occurs during normal exercise. All the mentioned studies use systemic Dysf knock-outs in adult mice, affecting the entire muscle at a stage when muscle stem cell pools have been used up and myopathic disease develops. Thereby, significant inflammation is triggered that leads to a massive immune response. It has been shown that, in these cases, even further damage is caused by the action of leukocytes. However, to the best of our knowledge, macrophages have not been shown to remove a protein lipid patch from a living cell that served to seal a membrane tear in a way that the cell can survive the injury. Our data show, for the first time, that cell membrane repair is a process that involves intrinsic mechanisms of a cell as well as cell-macrophage interactions.

Minor points:

1. Line 40: two periods after integrity

Corrected

2. Gene symbol: DYSF for humans (please correct in line 40)

Corrected

3. Line 45: What kind of multimeric complexes? Are they dependent or independent of Dysferlin?

This has been explained in Roostalu and Strähle Dev Cell 2012. Sorry, we cannot go into too many details for space considerations. We quote the original literature and excellent recent reviews for the uniformed reader. They comprise the annexins and are independent of Dysf.

4. Line 48: Knockdown is typically associated with gene and hence gene symbols should be used (italicized) or something like Down-regulation of Dysf or AnxA6 translation.

In this case we feel that our way of presenting is correct too. As we talk about protein translation and the way we have written Dysf and AnxA6 is correct for

zebrafish proteins according to ZFIN.

Similarly, Anxa2a-mO is non-italicized but Dys-mO is (lines 107-110).

Corrected

5. Line 97: Please define observation time

Done.

6. Reporter lines are described with italicized symbols in text but with non-italicized symbols in figure legends. These should be uniform (italicized).

We checked the manuscript very carefully including the legends. In most cases we refer to the fluorescent signal in the panels. Since the fluorescence signal is derived from the tagged proteins, the non-italicized way of writing is in our view correct. We corrected, however, the names of the morpholinos that should be in italics and some other errors in the nomenclature that the reviewer did not specifically point out.

Doherty, K. R., A. Cave, et al. (2005). "Normal myoblast fusion requires myoferlin." Development **132**(24): 5565-5575.

Segawa, K., S. Kurata, et al. (2014). "Caspase-mediated cleavage of phospholipid flippase for apoptotic phosphatidylserine exposure." Science **344**(6188): 1164-1168.

REVIEWERS' COMMENTS:

Reviewer #1 (Remarks to the Author):

The authors have addressed all my concerns.

Reviewer #2 (Remarks to the Author):

The authors responded well to my concerns. I have one more minor point.

Line 338, "The Ca²⁺ dependent caspases": caspase is NOT dependent on Ca²⁺. Reference 40 reports that caspase 3/7 in apoptotic cells or a high concentration of Ca²⁺ in activated platelets inhibits flippase. This sentence should be corrected.

Reviewer #3 (Remarks to the Author):

Authors have made significant changes to the first submission. However, following still need to be addressed

Major Points:

1. In line 54: authors point that none of the vesicles that were described in previous studies contributed to membrane repair in zebrafish myofibers. Could these differences be due to species related differences then scientific discrepancy. Authors need to explain that well.

Authors didn't find any involvement of vesicles in patch repair in zebrafish like previously seen in mammalian cells. As they pointed out, performing studies on mammalian cells is altogether another experimental study. However, they need to discuss the differences and potential reason for these differences in two systems. Proposed studies are performed with an implication in understanding human disease biology. Therefore, any differences in two model systems need to be addressed.

2. A single morpholino by itself can result in non-specific effects. The use of three morpholinos is a bit unusual for knockdown studies. Please provide the concentrations of these morpholinos and controls employed to rule out non-specific effects.

Authors show that the results obtained by morpholinos are reproducible in an *irf8* knockout model as well. They provided a figure in the response letter but have not added this in the MS. It will be helpful to add this even if as a supplemental figure.

3. Authors describe the use of dysferlin mutant. Has this mutant been published? Does this mutant show similar phenotypes as of morphant fish and/or human disease? Why did authors choose to work with morpholino while they already have a genetically clean mutant model available?

Authors mention that mutant recapitulates the morphant's phenotype of defect in repair patch. Has this been reported earlier (please provide the reference). Otherwise data needed to be added the manuscript supporting this statement.

4. Authors have used HeLa cells to confirm the conservation between zebrafish and human studies. It will be highly relevant to show these in muscle cells, primary myoblasts or differentiated cells.

Authors are requested to add C2C12 data to the manuscript as well. As most of the researchers used C2C12 cells/myofibers for dysferlin studies, it will be very valuable for others in this field.

Minor points:

1. Reporter lines are described with italicized symbols in text but with non-italicized symbols in figure legends. These should be uniform (italicized)

Gene/protein nomenclature needs to be checked carefully. Eg. Line 156 has *dysf*-MO while 158 has *Dysf*-MO. For mutant authors used *dysf*, line 162 (correct nomenclature) however, for morphants they still use protein nomenclature for morphants. When we talk about morphants its similar to mutants and therefore correct gene symbol should be used (e.g. line 160 *AnxA6* should have been *anxa6*). Protein symbols are depicted as *AnxA6*.

REVIEWERS' COMMENTS:

Reviewer #1 (Remarks to the Author):

The authors have addressed all my concerns.

Reviewer #2 (Remarks to the Author):

The authors responded well to my concerns. I have one more minor point.

Line 338, "The Ca²⁺ dependent caspases": caspase is NOT dependent on Ca²⁺. Reference 40 reports that caspase 3/7 in apoptotic cells or a high concentration of Ca²⁺ in activated platelets inhibits flippase. This sentence should be corrected.

The sentence has been corrected (MS line 351).

Reviewer #3 (Remarks to the Author):

Authors have made significant changes to the first submission. However, following still need to be addressed

Major Points:

1. In line 54: authors point that none of the vesicles that were described in previous studies contributed to membrane repair in zebrafish myofibers. Could these differences be due to species related differences then scientific discrepancy. Authors need to explain that well. Authors didn't find any involvement of vesicles in patch repair in zebrafish like previously seen in mammalian cells. As they pointed out, performing studies on mammalian cells is altogether another experimental study. However, they need to discuss the differences and potential reason for these differences in two systems. Proposed studies are performed with an implication in understanding human disease biology. Therefore, any differences in two model systems need to be addressed.

We appreciate the reviewer's interest in this question. However, expanding this topic would mean expanding on an issue that is definitely not in the main focus of the paper. In our opinion, an introduction should lead to the main theme of the results section of the paper. Moreover, it will not be possible to address this issue in a correct and precise way with a few sentences: The differences seen with respect to involvement of different vesicles in

membrane repair in different organisms and cell types may have many unrelated reasons and the biggest concern is that the differences arise from species and cell type differences as well as technical parameters. It is well known that the lesion size influences how the holes are repaired : e.g. toxin or otherwise induced pores in nanometer range are repaired with a different set of repair proteins compared to μm scale lesions, indicating that lesion size matters for repair. On the other hand smaller cells as e.g. HeLa cells might use a different set of repair proteins indicated e.g. by very low expression of Dysf, thereby varying the set of repair proteins from cell type to cell type. This variation will also include the set of vesicles or sources of repair proteins and lipids for membrane repair. As membrane repair is still poorly understood especially using comparative studies it is not known what exact role e.g. lysosomes play in membrane repair. They could have a role in membrane repair at later stages in the zebrafish as well ie in systemic expansion of the surface of the cell membrane. It is not clear whether or not in human cells lysosomes play a direct or indirect role. It is only known that they are in some way involved in membrane repair. Altogether, we believe that the better way is to stay with our solution of the problem when we state in the introduction: “Lysosomes as well as endosomes were suggested to contribute to membrane repair in mammalian cell systems^{1,2}. However, none of the tested intracellular vesicles marked by Laptm4a, Lamp1, Lamp2, Rab1a, Rab5a, Rab6a, Rab7 and Rab27a contribute significantly to repair patch formation in zebrafish myofibers³.”

2. A single morpholino by itself can result in non-specific effects. The use of three morpholinos is a bit unusual for knockdown studies. Please provide the concentrations of these morpholinos and controls employed to rule out non-specific effects. Authors show that the results obtained by morpholinos are reproducible in a irf8 knockout model as well. They provided a figure in the response letter but have not added this in the MS. It will be helpful to add this even if as a supplemental figure.

We are puzzled by this comment of the reviewer as we have addressed and explained this issue in all length in the rebuttal letter of the previous version and also included additional data in the previous response letter. See for example the response to this reviewer’s comment # 4 (another answer to this question was given in response to comment 2 of reviewer #2):

“Concentrations of morpholinos directed against pu.1 (final 0.5 mM); gcsfr (final 0.5 mM); irf8 (final 0.6 mM) are as given. We added this to the materials and methods section. Accordingly, the total concentration of morpholinos was 1.6 mM. The control was set to 1.6 mM as well. In order to check whether the triple MO knock-down influences the repair process, the relative fluorescence of AnxA2a-mO in the repair patch was measured and quantified (see below Fig. J). As can be seen from the data, the accumulation is normal and comparable to AnxA2a-mO accumulation shown in Figure 2a. Also, embryos developed normally and muscle birefringence is normal in triple morphants. Please also see our remarks to comment #2 of Reviewer 2.

Fig. J Accumulation of AnxA2a-mO at the repair patch over time in embryos in which macrophages were knocked-down (red) by triple injection of Mo-pu.1 (final 0.5 mM); MO-gcsfr (final 0.5 mM); MO-irf8 (final 0.6 mM) versus control MO injected zebrafish (green). 150% relative fluorescence was reached at ~155 s (control) and ~140 s (KD).

Irrespectively, we anticipated criticism along these lines and established an *irf8* knock-out line by CRISPR/Cas9 technology while the paper was under review. In contrast to the morpholinos, *irf8* single knock-out is sufficient to eliminate macrophages, at least up to the developmental stages that we used to analyse the

Fig. K Crispr/Cas9 knock-out of *Irf8* impairs membrane repair patch removal. In controls, the repair patch was removed in 32% of injured myofibers (n=19) within one hour after injury. In the *irf8*-KO embryos the patch was present in all cases of injured myofibers examined (n=15). Significance was checked with Fishers exact test at $p < 0.05$ ($p = 0.023895$ *).

repair of the membrane lesions. As a positive control, damage to several myofibers was induced and the removal of dead cells by macrophages was checked over 18 h. While in WT siblings, dead fibers had been removed (n=10 experiments), dead myofibers persisted in *irf8*-KO fish (n=10 experiments) over the whole imaging time of 18 h, as seen for the triple knock-down ($p = ***$). Thus, the requirement of a triple knock-down (see also answer to comment 2 of reviewer 2) is most likely due

to the leakiness of the individual morpholinos. We also checked the removal of repair patches in injured myofibers (Fig. K). The patches were removed in 32% of injured myofibers in wildtype siblings. In contrast, none of the patches were removed in *irf8* mutant embryos (Fig. K). These results confirm our findings of the triple knock-down by an independent and genetic method and fully support our conclusions. “

The results of the *irf8* knockout have now been included in the manuscript as Fig. 1e. We believe that our much more solid genetic control data are sufficient to make our point. The other two reviewers were fully satisfied with our answers.

3. Authors describe the use of dysferlin mutant. Has this mutant been published? Does this mutant show similar phenotypes as of morphant fish and or human disease? Why did authors chose to work with morpholino while they already have a genetically clean mutant model available?

We had answered this question already in the last rebuttal. This reviewer may have overseen these answers. We had answered in response to comment 6 of this reviewer: “The mutant has not yet been published. The mutant recapitulates the defect of the morphant in the repair patch, i.e., loss of PS accumulation. The mutant does not show the effect on muscle birefringence seen previously for the *dysf* morphants. This may be the result of maternal contribution, compensatory or unspecific effects. Currently, we also cannot exclude that this is a reflection of the slightly different strategies used to knock down or knock out *dysf*. The major goal for this study was to reproduce the morphant phenotype with respect to repair patch formation and PS accumulation. Further studies, also in adult zebrafish to assess the effects on muscle physiology and structure, are needed to fully describe the mutant and answer the questions of the reviewer. “

We had also referred the reviewer 3 to the answer to comment 7 of reviewer 1:

“Yes, we observe failure to transport PS efficiently to the lesion patch in the *dysf* morphants as well the *dysf* mutant, in line with the defective membrane repair ability in LGMD2B patients. In addition, we observed an effect on birefringence of the somitic musculature in the morphants, which is an indication of impaired myofibril organisation. This impaired myofibrillar organisation was, however, not observed in the mutant. The predominant purpose was to use the mutant to show that the effects that we see by knock-down of *dysf* expression in repair patch formation can be reproduced by a gene knock-out. We developed the genetic *dysf* knock-out only recently and, therefore, an in-depth analysis of the zebrafish mutant is not available yet. A detailed analysis, especially also of the adult musculature, has not been carried out yet and would be a considerable effort far beyond the scope of this manuscript. “

We have generated the *dysf* mutant at a late stage of this project when it had suddenly become clear that the standard controls for morpholinos may fail. As such

this mutant served primarily as a control of the effects seen with the *dysf* morpholino. As stated above and also in the manuscript, we have reproduced this effect.

Authors mention that mutant recapitulates the morphant's phenotype of defect in repair patch. Has this been reported earlier (please provide the reference). Otherwise data needed to be added the manuscript supporting this statement.

Now we are totally confused. We have clearly stated that we have generated the mutant and it has not been reported before. Even more so, *Dysf* as carrier of PS to the lesion has not been reported before either. This is one of the central findings of the paper. How can the reviewer ask us to quote the evidence for this, when the original evidence is provided in the paper under review? This is unquestioned by the other two reviewers.

4. Authors have used HeLa cells to confirm the conservation between zebrafish and human studies. It will be highly relevant to show these in muscle cells, primary myoblasts or differentiated cells.

*Authors are requested to add C2C12 data to the manuscript as well. As most of the researchers used C2C12 cells/myofibers for *dysferlin* studies, it will be very valuable for others in this field.*

We take the point to add the C2C12 data. These were presented (!) in the rebuttal letter of the previous version. The statement of the reviewer which makes things appear as if we had not done the experiments is utterly grotesque. The data were added as Supplementary Figures (Supplementary Fig. 6g-l) in the manuscript.

Minor points:

1. Reporter lines are described with italicized symbols in text but with non-italicized symbols in figure legends. These should be uniform (italicized)

*Gene/protein nomenclature needs to be checked carefully. Eg. Line 156 has *dysf*-MO while 158 has *Dysf*-MO. For mutant authors used *dysf*, line 162 (correct nomenclature) however, for morphants they still use protein nomenclature for morphants. When we talk about morphants its similar to mutants and therefore correct gene symbol should be used (e.g. line 160 *AnxA6* should have been *anxa6*). Protein symbols are depicted as *AnxA6*.*

We have carefully checked the manuscript. To our best knowledge, stylistic representation of names of genes, proteins, mRNAs, trasngenes and morpholinos are correct.

1. Idone V, Tam C, Goss JW, Toomre D, Pypaert M, Andrews NW. Repair of injured plasma membrane by rapid Ca²⁺-dependent endocytosis. *J Cell Biol* **180**, 905-914 (2008).
2. Reddy A, Caler EV, Andrews NW. Plasma membrane repair is mediated by Ca²⁺-regulated exocytosis of lysosomes. *Cell* **106**, 157-169 (2001).
3. Roostalu U, Strahle U. In vivo imaging of molecular interactions at damaged sarcolemma. *Dev Cell* **22**, 515-529 (2012).